# LEARNING TO ANSWER FROM CORRECT DEMONSTRATIONS

**Nirmit Joshi,**[*] **Gene Li**
Toyota Technological Institute at Chicago
nirmit@ttic.edu
gene@ttic.edu

**Siddharth Bhandari, Shiva Prasad Kasiviswanathan**
Amazon
siddharthbhandari.engg@gmail.com
kasivisw@gmail.com

**Cong Ma**
University of Chicago
Department of Statistics
congm@uchicago.edu

**Nathan Srebro**
Toyota Technological Institute at Chicago
nsrebro@ttic.edu

## ABSTRACT

We study the problem of learning to generate an answer (or completion) to a question (or prompt), where there could be multiple correct answers, any one of which is acceptable at test time. Learning is based on demonstrations of some correct answer to each training question, as in Supervised Fine Tuning (SFT). We formalize the problem as imitation learning (i.e., apprenticeship learning) in contextual bandits, with offline demonstrations from some expert (optimal, or very good) policy, without explicitly observed rewards. In contrast to prior work, which assumes the demonstrator belongs to a bounded-complexity policy class, we propose relying only on the underlying reward model (i.e., specifying which answers are correct) being in a bounded-complexity class, which we argue is a strictly weaker assumption. We show that likelihood-maximization methods can fail in this setting, and instead present an approach that learns to answer nearly as well as the demonstrator, with sample complexity logarithmic in the cardinality of the reward class. Our method is similar to Syed & Schapire (2007), when adapted to a contextual bandit (i.e., single step) setup, but is a simple one-pass online approach that enjoys an "optimistic rate" (i.e., $1/\varepsilon$ when the demonstrator is optimal, versus $1/\varepsilon^2$ in Syed & Schapire), and works even with arbitrarily adaptive demonstrations.

**Please see the full version at arXiv:2510.15464.**

## 1 INTRODUCTION

Many real-world problems involve generating an answer to a question, where there may be many *equally good* responses. A math question can have millions of equally valid but differently written solutions, a coding task can admit many different perfectly working implementations, and a recommendation query can be satisfied by multiple items. The learner's challenge is *not* to reproduce all correct responses, but to generate a *single good answer*.[1]

This recurring structure can be formalized as *contextual bandits*. The context and actions are the questions $x \in \mathcal{X}$ and answers $y \in \mathcal{Y}$, respectively. In the simplest case, consider an unknown binary reward function $r_\star(x, y) \in \{0, 1\}$ indicating whether a response is correct. We get demonstrations that are always correct, i.e., from a training set $S = \{(x_i, y_i)\}_i$ where $x_i \sim \mathcal{D}, y_i \sim \widetilde{\pi}(\cdot \mid x_i)$, where $\mathcal{D}$ is some context distribution and $\widetilde{\pi}(\cdot \mid x)$ is a demonstration policy supported on the set of *optimal* or *correct* actions for the context $x$, given by[2]

$$\sigma_\star(x) := \arg\max_{y \in \mathcal{Y}} \ r_\star(x, y) \underbrace{= \{y \in \mathcal{Y} : r_\star(x, y) = 1\}}_{\text{for 0-1 rewards with correct answers}} \tag{1}$$

---

[*]The work was done while NJ was interning at Amazon.

[1]Throughout, we use "good" and "correct" interchangeably. One should think of a "correct" answer in this context as a "full credit complete answer", not merely one that is "logically true".

[2]In the special case of binary rewards with always correct demonstrations, we are also implicitly assuming every question has at least one correct answer. This is not required in the more general treatment.

Note that $|\sigma_\star(x)|$ can be huge and there may be many correct answers.

More generally, we consider real-valued rewards $r_\star(x, y) \in [0, 1]$, and do not require the demonstrator $\widetilde{\pi}$ to be optimal (i.e., to always answer in $\sigma_\star(x)$). Our goal is to compete with the *value* of the demonstrator, where the value of a policy $\pi : \mathcal{X} \to \Delta(\mathcal{Y})$ is defined as:

$$V_{r_\star}(\pi) = \mathbb{E}_{x \sim \mathcal{D}} \mathbb{E}_{y \sim \pi(\cdot|x)}[r_\star(x, y)]. \tag{2}$$

For binary rewards, $V_{r_\star}(\pi)$ is exactly the *accuracy* of the predictor $\pi$. Our goal is to learn a predictor $\widehat{\pi}$ that produces *good* responses to unseen contexts sampled from $\mathcal{D}$, as captured by $V_{r_\star}(\widehat{\pi})$. In particular, we would like to learn a predictor whose value (i.e., accuracy) is nearly as good as that of the demonstrator:

$$V_{r_\star}(\widehat{\pi}) \geq V_{r_\star}(\widetilde{\pi}) - \varepsilon. \tag{3}$$

This is *apprenticeship learning* (Abbeel & Ng, 2004; Syed & Schapire, 2007), aka *imitation learning*, in contextual bandits from offline data. We argue that in modern large language model (LLM) training, this is exactly the problem addressed during the *supervised fine-tuning* (SFT) phase,[3] where the model is trained on questions with expert-generated good (perhaps near-optimal) answers, and the goal is to be able to answer *as well as* the expert.

An important aspect of our objective in Eq. (2) is that it is entirely reward-driven, focusing solely on *utility*. We do not require matching the distribution of $\widetilde{\pi}$. In most question–answering or prompt-response systems, we argue that the true objective is often reward maximization rather than distribution matching. For example, in tasks such as producing gold-medal-winning IMO solutions, the solution set $\sigma_\star(\text{question}) = \{\text{all completely correct solution texts}\}$ (i.e., reward = 1), is unfathomably large, due both to variation in solution approaches and details, and to differences in the solution text itself, down to word choices, spacing, and punctuation. Rather than matching the generation style of a particular expert, or some arbitrary distribution over experts, producing any single correct solution is sufficient to win a gold medal. The same principle underlies the deployment of modern LLMs, where success is judged by whether the model generates a single high-utility answer to the user's question (OpenAI, 2023; Anthropic, 2024; DeepMind, 2023; DeepSeekAI, 2025; Touvron et al., 2023).

**Demonstrator Class Assumption and Likelihood Maximization.** A common approach when learning from demonstrations is to assume that the demonstrator $\widetilde{\pi}$ belongs to a low-capacity policy class $\Pi$. This motivates maximum likelihood estimation (MLE), or equivalently log-loss minimization, as a natural learning rule; see, e.g., Foster et al. (2024). Indeed, in SFT for LLMs, one fits a model by minimizing log-loss on prompt–completion pairs. Foster et al. (2024)—relying on the fact that MLE enjoys sharp $O(\frac{\log |\Pi|}{m})$ convergence guarantees in distribution from $m$ samples under a certain notion of distance—also enjoys low reward suboptimality, with tight rates for the value suboptimality in (2) (see Proposition 1 in Section 3, based on Foster et al., 2024). That is, not only does MLE achieve small value suboptimality; it does so by *cloning* the demonstrator's action distribution.

However, for this guarantee to be meaningful, $\log |\Pi|$ must be small, and the demonstrator must lie in this class. This requires strong assumptions about the exact behavior of the demonstrator—for example, modeling the generative process of specific students producing solutions to math problems, rather than just modeling what it means for a solution to be correct.

**Reward Class Assumption and Our Results.** Instead, we propose modeling the rewards themselves rather than the demonstrator policy class, and rely only on the reward class having low cardinality (Section 2). Concretely, we assume the unknown reward function $r_\star : \mathcal{X} \times \mathcal{Y} \to [0, 1]$, specifying the utility of a particular answer for a given question, comes from a known low-cardinality reward class $\mathcal{R} \subseteq [0, 1]^{\mathcal{X} \times \mathcal{Y}}$.

---

[3]Response generation by LLMs can be seen as a Markov Decision Process (MDP), where the response $y$ is a token sequence and each token is an action. In this MDP, transitions are deterministic, the 'state' includes the full prompt $x$ and all previously generated tokens, and rewards are sparse—typically only given at the end. Thus we find treating response generation as a contextual bandit with the entire response $y$ as a single action, as a more useful abstraction for our purposes. Indeed, treating $y$ as a single action is also the view routinely taken in the study of question-answering for LLMs (e.g., Rafailov et al., 2023; Huang et al., 2025; Wu et al., 2025, etc.). See also Remark 3.

| Assump.
Rule | low-cardinality $\Pi$ | low-cardinality $\mathcal{R}$ |
|---|---|---|
| MLE | $\sqrt{\dfrac{\Delta \log \|\Pi\|}{m}} + \dfrac{\log \|\Pi\|}{m}$
Proposition 2 | May not learn

Theorems 1 and 2 |
| Our Learner | $\dfrac{\log \|\Pi\|}{m}$
Corollary 2 | $\sqrt{\dfrac{\Delta \log \|\mathcal{R}\|}{m}} + \dfrac{\log \|\mathcal{R}\|}{m}$
Theorem 5 |

Table 1: The case of optimal demonstrations provides a link between the low-cardinality assumptions on $\Pi$ and $\mathcal{R}$. The black text compares the sample complexity of MLE and our learner under both assumptions when the demonstrator is optimal, establishing that a low-cardinality assumption on $\mathcal{R}$ is strictly weaker than low-cardinality $\Pi$. The slower decaying terms for both learners under their respective assumptions are shown in gray; these interpolate between the slow $1/\sqrt{m}$ and the fast $1/m$ rates depending on the demonstrator suboptimality $\Delta$.

For example, one can think of the reward given by an encoder-only transformer that takes the sequences $x$ and $y$ as input and outputs a single real-valued reward. In this view, the class $\mathcal{R}$ corresponds to all possible transformers obtained by varying the model's weights. To simplify analysis and presentation, we consider finite cardinality classes $\mathcal{R}$ and rely on the log-cardinality $\log \|\mathcal{R}\|$. The log-cardinality should be thought of as corresponding to the number of parameters—e.g., if the transformer has $N$ parameters, each a 32-bit float, then $\log \|\mathcal{R}\| \leq 32N$.

We show that the reward class assumption is, in a sense, a weaker assumption than the demonstrator class assumption, and that it motivates looking beyond likelihood maximization/log-loss minimization and suggests different learning procedures:

- In Section 2, we show that **assuming the reward lies in a low-cardinality class is a strictly weaker assumption**, at least when the demonstrator $\widetilde{\pi}$ is optimal for the unknown reward $r_\star$, compared to assuming the demonstrator lies in a low-cardinality policy class.

- In Section 3, we show that **MLE can fail to generalize for low-cardinality reward classes** in simple situations (Theorems 1 and 2).

- In Section 4, we present a **learner that succeeds with optimal** $O(\log \|\mathcal{R}\|/m)$ **samples in competing with an optimal demonstrator**, and this sample complexity degrades gracefully when the demonstrator is suboptimal via an "optimistic rate" (Theorem 5). The guarantee has no dependence on the size $\|\mathcal{Y}\|$, or the size of the optimal action sets $\|\sigma_\star(x)\|$.

See Table 1 for a comparison between MLE and our learner under low-cardinality assumptions on $\Pi$ and $\mathcal{R}$. In Appendix A.3, we discuss the important distinction between distribution matching (which MLE attempts to achieve) and alternative approaches (like ours) that do not rely on distribution matching to obtain good performance. In particular, distribution matching is impossible under the Reward Class Assumption; thus, any approach that ensures low value-suboptimality must do so without relying on distribution matching (unlike MLE). Finally, in Appendix F, we also discuss a pass@$k$ extension of our learner, which is minimax optimal for the pass@$k$ metric when the demonstrations are optimal.

**Relationship to Syed & Schapire (2007).** While preparing this manuscript, we learned of this related work in the more general setting of *known*, and (possibly) *stochastic* transitions. The method and analysis of Syed & Schapire (2007) can be adapted to contextual bandits, yielding a method similar to ours, and already guaranteeing the reward suboptimality of $\sqrt{\frac{\log \|\mathcal{R}\|}{m}}$. We provide a correspondence in Appendix A.2.

In this paper, we emphasize the application to contextual bandits, or equivalently in the known model class of MDPs with *deterministic transitions* (see Remark 3), and contrast it with policy-class-based likelihood maximization methods. In our setting, our method enjoys an optimistic rate (as in Table 2), and allows us to argue strict dominance over MLE in the case of an optimal demonstrator. Our result is derived from a regret-minimization algorithm for the online setup with an *adversar-*

*ial* sequence of demonstrations (see Figure 1). Additionally, in the statistical setting, our learning algorithm is a one-pass online algorithm, as opposed to a multi-pass batch approach, with a self-contained and simpler analysis of fast rates; with the faster $1/\varepsilon$ rate holding for arbitrarily adaptive but *good* demonstrations.

## 2   SETTING AND PROBLEM DEFINITION

Let $\mathcal{X}$ and $\mathcal{Y}$ respectively be the sets of all possible contexts and actions. A *policy* (or *predictor*, we use the words interchangeably) $\pi$ is specified by a mapping $x \mapsto \pi(\cdot \mid x)$ from contexts to distributions over actions. We overload notation and view a policy as a randomized mapping, so $\pi(x)$ denotes a random variable. If $\pi$ is deterministic, $\pi(x)$ is the action taken for the context $x$. Recall the definition of the value of a policy $\pi$ for a reward $r$ from Eq. (2), given by $V_r(\pi) = \mathbb{E}_{x \sim \mathcal{D}} \mathbb{E}_{y \sim \pi(\cdot \mid x)}[r(x, y)]$. The value also depends on the unknown population distribution $\mathcal{D}$ over context, which is implicit in the notation. We use $(x, y) \sim \mathcal{D} \times \pi$ to denote a joint distribution where $x \sim \mathcal{D}$ and $y \sim \pi(\cdot \mid x)$. Learning is based on an i.i.d. samples from $\mathcal{D} \times \widetilde{\pi}$, for some demonstrator policy $\widetilde{\pi}$. The goal is to always compete with the value of the demonstrator $V_{r_\star}(\widetilde{\pi})$ w.r.t. the ground truth reward function $r_\star$. This is formalized in Appendix A.1.

Crucial to our work is the distinction between two types of hypothesis classes and corresponding realizability assumptions:

**Reward Class Assumption.** The unknown reward function $r_\star$ is in a known hypothesis class of reward functions, $r_\star \in \mathcal{R} \subseteq [0, 1]^{\mathcal{X} \times \mathcal{Y}}$. The demonstrator policy $\widetilde{\pi}$ is arbitrary.

**Demonstrator Class Assumption.** The unknown demonstrator $\widetilde{\pi}$ is in a known demonstrator policy class $\widetilde{\pi} \in \Pi \subseteq (\Delta(\mathcal{Y}))^{\mathcal{X}}$. The reward function $r_\star : \mathcal{X} \times \mathcal{Y} \to [0, 1]$ is arbitrary.

## 3   WHY MLE IS GOOD FOR LOW-CARDINALITY $\Pi$ BUT FAILS FOR $\mathcal{R}$?

Given a policy class $\Pi \subseteq (\Delta(\mathcal{Y}))^{\mathcal{X}}$, the (conditional) Maximum Likelihood Estimator (MLE) (or the log-loss minimizer) is defined as

$$\mathrm{MLE}_\Pi(S) := \arg\max_{\pi \in \Pi} \prod_{i=1}^{m} \pi(y_i \mid x_i) = \arg\min_{\pi \in \Pi} \left( -\sum_{i=1}^{m} \log \pi(y_i \mid x_i) \right). \tag{MLE}$$

### 3.1   LEARNING UNDER THE DEMONSTRATOR CLASS ASSUMPTION

To contrast with our work, we first consider learning under the Demonstrator Class Assumption, where the demonstrator class $\Pi$ has small cardinality. In this setting, MLE is *minimax optimal*. The argument uses that MLE achieves an $O(\frac{\log |\Pi|}{m})$ convergence rate in squared Hellinger distance to the demonstrator's distribution (see Proposition 2), where

$$D_{\mathsf{H}}^2(\mathbb{P}, \mathbb{Q}) := \sum_{z \in \mathcal{Z}} \left( \sqrt{\mathbb{P}(z)} - \sqrt{\mathbb{Q}(z)} \right)^2$$

for distributions $\mathbb{P}, \mathbb{Q}$ over a discrete domain $\mathcal{Z}$. A simple change-of-measure argument then shows that, when the demonstrator is *optimal*, the value suboptimality is also bounded by $D_{\mathsf{H}}^2(\mathbb{P}, \mathbb{Q})$, and hence scales as $O(\frac{\log |\Pi|}{m})$. In any case, the value suboptimality can be bounded by a total variation distance, which decays as $O(\sqrt{\frac{\log |\Pi|}{m}})$, since $D_{\mathsf{TV}}(\mathbb{P}, \mathbb{Q}) \leq \sqrt{D_{\mathsf{H}}^2(\mathbb{P}, \mathbb{Q})}$. More generally, one can show the following proposition, with rates that interpolate between the two (in a similar spirit to Foster et al. (2024), and proved for completeness in Appendix D.1):

**Proposition 1.** *For any finite $\Pi \subseteq (\Delta(\mathcal{Y}))^{\mathcal{X}}$ where $(\mathcal{X}, \mathcal{Y})$ are countable, for any $\mathcal{D}$, any $\widetilde{\pi} \in \Pi$, with probability at least $1 - \delta$ over $S \sim (\mathcal{D} \times \widetilde{\pi})^m$, any $\widehat{\pi}_{\mathrm{mle}} \in \mathrm{MLE}_\Pi(S)$ enjoys the following guarantee:*
*(1) For any $r_\star : \mathcal{X} \times \mathcal{Y} \to [0, 1]$ for which $\widetilde{\pi}$ is optimal, we have*

$$V_{r_\star}(\widetilde{\pi}) - V_{r_\star}(\widehat{\pi}_{\mathrm{mle}}) \leq D_{\mathsf{H}}^2(\mathcal{D} \times \widehat{\pi}_{\mathrm{mle}}, \mathcal{D} \times \widetilde{\pi}) \leq \frac{6 \log(2|\Pi|/\delta)}{m}.$$

*(2) For any $r_\star : \mathcal{X} \times \mathcal{Y} \to [0, 1]$, we have*

$$V_{r_\star}(\widetilde{\pi}) - V_{r_\star}(\widehat{\pi}_{\mathrm{mle}}) \leq D_{\mathsf{TV}}(\mathcal{D} \times \leq \widehat{\pi}_{\mathrm{mle}}, \mathcal{D} \times \widetilde{\pi}) \leq \sqrt{\frac{6 \log(2|\Pi|/\delta)}{m}} \, .$$

*(3) More generally, for any $r_\star : \mathcal{X} \times \mathcal{Y} \to [0, 1]$, we have*

$$V_{r_\star}(\widetilde{\pi}) - V_{r_\star}(\widehat{\pi}_{\mathrm{mle}}) \leq O\left( \sqrt{\frac{\Delta_{\widetilde{\pi}} \ \log(|\Pi|/\delta)}{m}} + \frac{\log(|\Pi|/\delta) \log m}{m} \right) ,$$

*where $\Delta_{\widetilde{\pi}} := \mathbb{E}_{(x,y) \sim \mathcal{D} \times \widetilde{\pi}}\left[ \sup_{y'} r_\star(x, y') - r_\star(x, y) \right]$ is the value suboptimality of the demonstrator.*

The proof follows ideas from Foster et al. (2024). We first use standard density-estimation guarantees to obtain convergence in squared Hellinger distance $D_{\mathsf{H}}^2(\mathcal{D} \times \widehat{\pi}_{\mathrm{mle}}, \mathcal{D} \times \widetilde{\pi})$ (see Proposition 2). A change-of-measure argument then shows that, under optimality, the value suboptimality is controlled by the squared Hellinger distance (and, in any case, by the total variation distance). This immediately implies the first two parts of the proposition. To obtain the tight interpolating rates, we rely on a variance-dependent decomposition of the suboptimality in terms of the squared Hellinger distance established by Foster et al. (2024); see Appendix D.1 for details.

## 3.2 Learning Under the Reward Class Assumption.

We now ask what can be said about the performance of the Maximum Likelihood Estimator in terms of the log-cardinality of the reward class $\mathcal{R}$. We show that even when $\mathcal{R} \subseteq \{0, 1\}^{\mathcal{X} \times \mathcal{Y}}$ is a binary reward class and the demonstrator is always correct, i.e., a demonstrator with value 1, the MLE still fails to learn. We will specify a class $\mathcal{R}$ such that the correct answers always exist, i.e., $\forall_{r \in \mathcal{R}, x \in \mathcal{X}} \exists_{y \in \mathcal{Y}}$ s.t. $r(x, y) = 1$.

First, since our problem is only specified in terms of the reward class $\mathcal{R}$, we need to specify which policy class $\Pi$ we want to maximize the likelihood over. Since we are promised that the demonstrator is optimal, the most natural choice is to consider the class $\Pi_{\mathcal{R}}$ from Eq. (14) of all policies supported on the correct answers for the rewards in $\mathcal{R}$. Recall that Reward Class Assumption with $\mathcal{R}$, along with optimality of the demonstrator, implies the Demonstrator Class Assumption with $\Pi_{\mathcal{R}}$. As discussed in Section 2, even if $\mathcal{R}$ is small, the class $|\Pi_{\mathcal{R}}|$ may be infinite (as long as there are even two correct actions for a reward). This makes the guarantee in Proposition 1 vacuous. Indeed, we will see that $\mathrm{MLE}_{\Pi_{\mathcal{R}}}$ can fail to learn with any sample size, even when $|\mathcal{R}| = |\mathcal{Y}| = 2$.

Let us think of how $\widehat{\pi}_{\mathrm{mle}} \in \mathrm{MLE}_{\Pi_{\mathcal{R}}}(S)$ would behave. For any $x$ observed in the training set, any distribution over correct answers is allowed, and only correct answers are observed, so $\widehat{\pi}_{\mathrm{mle}}(\cdot \mid x)$ will match the empirical distribution $\pi_S(\cdot|x)$ of observed $y_i$'s for $x_i = x$. However, on unseen contexts $x$, it may choose any distribution over actions $y \in \sigma_r(x)$ for some $r$ that is consistent with the data, i.e., in:

$$\mathrm{CONS}_{\mathcal{R}}(S) = \left\{ r \in \mathcal{R} \mid \forall_{(x,y) \in S} \ r(x, y) = 1 \right\}. \tag{4}$$

And the MLE predictor can be written as:

$$\widehat{\pi}_{\mathrm{mle}}(\cdot|x) = \begin{cases} \pi_S(\cdot|x), & \text{if } x \text{ is observed}; \\ \text{any distribution with support} \subseteq \sigma_r(x), & \text{otherwise}. \end{cases} \quad \text{for some } r \in \mathrm{CONS}_{\mathcal{R}}(S). \tag{5}$$

Since the loss on observed contexts is zero, the loss of the MLE predictor $\widehat{\pi}_{\mathrm{mle}}$ is exactly the same as just choosing any $r \in \mathrm{CONS}_{\mathcal{R}}(S)$ consistent with the data, and predicting actions $y \in \sigma_r(x)$ on any unseen context $x$. Maximizing the likelihood over a policy class does not buy us anything for unseen contexts over just ensuring consistency. While relying on just consistency is fine, even optimal, in (realizable, finite cardinality) supervised learning, we will see that it fails for our problem, and so does MLE which is equivalent to it.

Consider only binary response space $\mathcal{Y} = \{0, 1\}$ and a reward class $\mathcal{R} = \{r_0, r_{01}\}$. For every $x \in \mathcal{X}$, $r_0$ considers only 0 as the correct answer (i.e., $\forall_x r_0(x, y) = 1 - y$ and thus only $\sigma_{r_0}(x) = \{0\}$). However, $r_{01}$ considers both $\{0, 1\}$ as the correct responses (i.e., $\forall_x r(x, 0) = r(x, 1) = 1$). If the true hypothesis is $r_\star = r_0$, then all observed labels are 0 from an optimal demonstrator. However, $r_{01}$ also remains consistent, and thus $\mathrm{MLE}_{\Pi_{\mathcal{R}}}(S)$ may output 1 at test time—*failing to generalize*

*and go beyond memorization*. Thus, for an input distribution $\mathcal{D}$ such that the *missing mass* (i.e., unobserved contexts) is arbitrarily close to 1, we get the following failure.

**Theorem 1** (Failure of MLE over $\Pi_{\mathcal{R}}$)**.** *There exists $\mathcal{R} \subseteq \{0,1\}^{\mathcal{X} \times \mathcal{Y}}$ with $|\mathcal{R}| = |\mathcal{Y}| = 2, \mathcal{X} = \mathbb{N}$, and a choice of $r_\star \in \mathcal{R}$ and $\widetilde{\pi}$ such that for every sample size $m$ and $\epsilon \in (0,1)$, there exists a marginal distribution $\mathcal{D}$ such that $V_{r_\star}(\widetilde{\pi}) = 1$ but almost surely for $S \sim (\mathcal{D} \times \widetilde{\pi})^m$, some $\widehat{\pi}_{\mathrm{mle}} \in \mathrm{MLE}_{\Pi_{\mathcal{R}}}(S)$ has the following guarantee:*

$$V_{r_\star}(\widehat{\pi}_{\mathrm{mle}}) \leq \epsilon \,.$$

The failure of $\mathrm{MLE}_{\Pi_{\mathcal{R}}}$ can be attributed to the fact that the induced demonstrator class $\Pi_{\mathcal{R}}$ is too large, with infinitely many hypothesis, even though $\mathcal{R}$ is small. We might instead consider using Maximum Likelihood Estimation, but over a smaller policy class whose size matches that of $\mathcal{R}$. A natural choice for such a policy class is:

$$\overline{\Pi}_{\mathcal{R}} := \{\overline{\pi}_r \mid r \in \mathcal{R}\} \quad \text{where} \quad \overline{\pi}_r(\cdot \mid x) = \mathrm{Unif}(\sigma_r(x)),$$

where for simplicity we consider $\sigma_r(x)$, the set of optimal or correct actions, to be finite here. The advantage is that we now have $|\overline{\Pi}_{\mathcal{R}}| = |\mathcal{R}|$. However, this policy class is *misspecified* in that the actual (and optimal) demonstrator $\widetilde{\pi}$ need not be uniform on all correct answers and so $\overline{\Pi}_{\mathcal{R}}$ might not contain $\widetilde{\pi}$. We again cannot rely on Proposition 1.

**Remark 1** (Overlap of MLE)**.** Interestingly, despite the misspecification, for any $r_\star \in \mathcal{R}$ and with any optimal demonstrator $\widetilde{\pi}$ (even if $\widetilde{\pi} \in \overline{\Pi}_{\mathcal{R}}$, i.e., under the Reward Class and Optimality assumptions), we can at least ensure that $\widehat{\pi}_{\mathrm{mle}} \in \mathrm{MLE}_{\overline{\Pi}_{\mathcal{R}}}(S)$ achieves a non-trivial overlap with the optimal (i.e., correct) answers $\sigma_\star(x)$. More specifically, Theorem 6 in Appendix D.3 ensures that for $S \sim (\mathcal{D} \times \widetilde{\pi})^m$ such that $\widetilde{\pi}$ acts optimally w.r.t. $r_\star \in \mathcal{R}$, any $\widehat{\pi}_{\mathrm{mle}} \in \mathrm{MLE}_{\overline{\Pi}_{\mathcal{R}}}(S)$,

$$\mathbb{P}_S \left( \mathbb{P}_{x \sim \mathcal{D}} \left[ \exists_{y \in \sigma_\star(x)} \, \widehat{\pi}_{\mathrm{mle}}(y|x) \geq \frac{1}{\kappa} \right] \geq 1 - \varepsilon \right) \geq 1 - \delta \,, \quad \text{for any} \quad m \geq \frac{1}{\varepsilon}\big(\log|\mathcal{R}| + \log(1/\delta)\big), \tag{6}$$

where $\kappa := \sup_{r,x} |\sigma_r(x)|$ are the maximum support size.

The guarantee in Eq.(6) says that the MLE generates a good response under $r_\star$ with probability at least $1/\kappa = 1/\sup_{r,x} \sigma_r(x)$. However, such an overlap is not our goal and it is not very helpful in general: The support sizes $\kappa$ can be huge (e.g., for essay answers with even minimal language variability, it is exponential in the length of the answer), and the error of $\widehat{\pi}_{\mathrm{mle}}$ can be as high as $1 - 1/\kappa$. One can show this is indeed the case—with large support sizes, the value can be arbitrarily close to zero, as formalized below and proved in Appendix D.2:

**Theorem 2** (Failure of MLE over $\overline{\Pi}_{\mathcal{R}}$)**.** *For any $\epsilon \in (0,1)$, there exists $\mathcal{R} \subseteq \{0,1\}^{\mathcal{X} \times \mathcal{Y}}$ with $|\mathcal{R}| = 2, |\mathcal{X}| = 1, |\mathcal{Y}| = 2\lceil 1/\epsilon \rceil$, such that for some choice of $\mathcal{D} \times \widetilde{\pi}$ and $r_\star \in \mathcal{R}$ such that $V_{r_\star}(\widetilde{\pi}) = 1$, for every sample size $m$, and almost surely for $S \sim (\mathcal{D} \times \widetilde{\pi})^m$, there is a unique $\widehat{\pi}_{\mathrm{mle}} \in \mathrm{MLE}_{\overline{\Pi}_{\mathcal{R}}}(S)$ and it has the value:*

$$V_{r_\star}(\widehat{\pi}_{\mathrm{mle}}) \leq \epsilon \,.$$

Theorems 1 and 2 establish that natural choices of MLE fail to learn simple binary reward classes $\mathcal{R}$ (cf. Definition 1) even from an always correct demonstrator, i.e., with value 1.

## 4 LEARNING ALGORITHMS

In this section, we present our main result: a learning rule that can learn from demonstrations (as in Definition 1) with tight optimal sample complexities with optimistic rates. To achieve this, it is useful to focus on an adversarial online version of learning from demonstrations. For conceptual simplicity, we first discuss the case of binary rewards with an always correct demonstrator. We present a logarithmic $O(\log |\mathcal{R}|)$ mistake online learner in Section 4.2. In Section 4.3 we generalize this to an online regret bound for real-valued rewards with arbitrary (not necessarily optimal) demonstrations. The desired statistical learner is then obtained using an online-to-batch conversion in Section 4.4.

---

**Online (Mistake-Unaware) Learning Process for Contextual Bandits:**

At each round $t = 1, 2, \ldots, T$:

1. The learner **receives** an instance $x_t$.

2. The learner **outputs** a response $\widehat{y}_t$ (based on past instances and demonstrations).

3. In the case of 0-1 rewards with always correct demonstrator: the learner is said to make a **mistake** iff $r_\star(x, \widehat{y}_t) \neq 1$. More generally, the learner is said to have received a reward $r_\star(x_t, \widehat{y}_t)$. Importantly, the learner does not know whether a mistake occurred, or generally, does not observe the obtained reward.

4. The learner **receives** a demonstration $y_t$. In the case of 0-1 rewards with always correct demonstrator, we are promised that $r_\star(x_t, y_t) = 1$. Generally, $y_t$ can be arbitrary.

---

Figure 1: The online learning process studied. In contrast to the standard online contextual bandit setup, in our setup the learner does *not* know whether it made a mistake (unobserved rewards), but *does* always receive some other (unrelated) demonstration. The demonstration is always promised to be correct in the special case considered in Sections 4.1 and 4.2. In the most general case studied in Section 4.3, the demonstrations can be arbitrary.

## 4.1 ONLINE SETTING

To make progress, we turn our attention to the even more challenging online version of the problem. In the online process, summarized in Figure 1, the instances $x_t \in \mathcal{X}$ are received sequentially, and on each instance the learner first responds with $\widehat{y}_t$ and then receives an arbitrary demonstration $y_t$. We will return to the general reward case later in Section 4.3, but for now, let us focus on the case of binary rewards, with always correct demonstrations $y_t$ s.t. $r_\star(x_t, y_t) = 1$ for some unknown reward function $r_\star \in \mathcal{R}$.

The goal of the learner is to output $\widehat{y}_t$ such that $r(x_t, \widehat{y}_t) = 1$, and we say that if $r_\star(x_t, \widehat{y}_t) \neq 1$, then the learner made a *mistake* at round $t$. Importantly, the learner *does not know it has made a mistake*—the learner receives no direct feedback on $\widehat{y}_t$ (unlike in a traditional contextual bandit setup, e.g., Langford & Zhang, 2007; Dudík et al., 2011; Agarwal et al., 2014), only some (possibly unrelated) correct answer $y_t$. Nevertheless, we want to ensure that for any $r_\star \in \mathcal{R}$, and any sequence of instances $x_t$ and correct demonstrations with $r_\star(x_t, y_t) = 1$, the learner will only make few mistakes with respect to $r_\star$.

## 4.2 LOGARITHMIC MISTAKE ONLINE LEARNER WITH ALWAYS CORRECT DEMONSTRATIONS

We now present Algorithm 1, which we will prove makes at most $\log |\mathcal{R}|$ mistakes. For now let us focus on binary rewards where the demonstrator is always correct, and setting the hyperparameter $\gamma = 1$ (see Section 4.3 for the general version). The algorithm maintains weights $w^{(t)}(r)$, which are initialized to be uniform over all $r \in \mathcal{R}$. Predictions are based on a weighted average of the reward. For binary rewards, this corresponds to outputting the response $\widehat{y}_t$ which maximizes the sum of the weights of all rewards $r$ under which it is a correct response to $x_t$ (i.e., s.t. $r(x_t, y_t) = 1$). This is similar to $\widehat{\pi}_{\text{Maj}}$. As with $\widehat{\pi}_{\text{Maj}}$, we zero out the weights of $r$ that are not consistent with the demonstration $y_t$, and thus completely remove them from consideration. The difference is that in addition, we also *increase* the weights of $r$ under which the predicted action $\widehat{y}_t$ is *not correct* (i.e., $r(x_t, \widehat{y}_t) \neq 1$, and so $r$'s vote did not help elect $\widehat{y}_t$). We do so despite the fact that we do not know whether $\widehat{y}_t$ is correct or not. It turns out that this up-weighting is sufficient to exponentially reduce the number of mistakes the algorithm makes:

**Theorem 3** (Online Guarantee). *On any sequence $((x_t, y_t))_{t \in \mathbb{N}}$ that has only correct demonstrations with respect to some $r \in \mathcal{R}$ (i.e., s.t. $\forall_t \ r(x_t, y_t) = 1$), Algorithm 1 with $\gamma = 1$ makes at most $\log |\mathcal{R}|$ mistakes with respect to $r$.*

See Section E.1 for the proof. We can use an online-to-batch conversion, as in Algorithm 2, to obtain a guarantee in the statistical setting of Definition 1 with sample complexity

---

**Algorithm 1** Online MISTAKE-UNAWARE-WEIGHT-UPDATE Rule

---

**Input:** A finite reward $\mathcal{R} \subseteq [0,1]^{\mathcal{X} \times \mathcal{Y}}$ and a hyperparameter $\gamma \in [0,1]$.

- Initialize $w^{(1)}(r) = 1$ for all $r \in \mathcal{R}$.
- In every round,
    1. **Receive** $x_t$.
    2. **Output** $\widehat{y}_t := \arg\max_{y \in \mathcal{Y}} \sum_{r \in \mathcal{R}} w^{(t)}(r) \, r(x_t, y)$.[4]
       So using the policy $\widehat{\pi}_t(x) := \arg\max_{y \in \mathcal{Y}} \sum_{r \in \mathcal{R}} w^{(t)}(r) \, r(x, y)$, for the input $x_t$.
    3. **Receive** $y_t$.
    4. In the always correct demonstrator case (when $\gamma = 1$), update for each $r \in \mathcal{R}$:

$$w^{(t+1)}(r) \leftarrow \begin{cases} 0 & \text{if } r(x_t, y_t) \neq 1\,; \\ w^{(t)}(r) & \text{if both } r(x_t, y_t) = r(x_t, \widehat{y}_t) = 1\,; \\ 2\,w^{(t)}(r) & \text{if } r(x_t, y_t) = 1 \text{ but } r(x_t, \widehat{y}_t) \neq 1\,. \end{cases} \tag{7}$$

More generally, update:

$$w^{(t+1)}(r) \leftarrow w^{(t)}(r)\,(1+\gamma)^{r(x_t,*)-r(x_t,\widehat{y}_t)}\,(1-\gamma)^{r(x_t,*)-r(x_t,y_t)}, \tag{8}$$

where $r(x, *) = \sup_y r(x, y)$.

---

$O\!\left(\varepsilon^{-1}\big(\log|\mathcal{R}| + \log\delta^{-1}\big)\right)$. However, rather than carrying this out formally at this point, we first generalize the online analysis beyond binary rewards with correct demonstrations, and then present a unified online-to-batch conversion and statistical guarantee.

### 4.3 REGRET GUARANTEE FOR BOUNDED REWARDS AND ARBITRARY DEMONSTRATIONS

Still sticking to the online model, we now discuss the most general setting and provide a guarantee for Algorithm 1. This generalizes the earlier analysis in two ways: (i) it handles real-valued bounded rewards, and (ii) it accounts for suboptimal demonstrators. We refer to Figure 1 for the general online learning process from demonstrations: after outputting $\widehat{y}_t$ for a context $x_t$, the learner is said to have received reward $r(x_t, \widehat{y}_t)$, which is not observed. The demonstrations $y_t$ may be arbitrary and are not guaranteed to be optimal. Accordingly, instead of counting "mistakes," we track total accumulated reward. For a fixed reward $r \in \mathcal{R}$, consider the total reward collected by the algorithm and by the demonstrator, as well as the optimal total reward achievable on the given sequence of contexts with respect to $r$, i.e., by playing an optimal action in each round:

$$\widehat{J}_r(T) := \sum_{t=1}^{T} r(x_t, \widehat{y}_t), \quad \widetilde{J}_r(T) := \sum_{t=1}^{T} r(x_t, y_t), \quad J_r^*(T) := \sum_{t=1}^{T} \sup_y r(x_t, y). \tag{9}$$

Since the demonstrator may no longer be optimal, we cannot completely exclude from consideration reward models for which a given demonstration is not optimal. Instead of the hard update rule in (7), we use the softer variant in (8): we again decrease the weight of reward functions $r$ for which the demonstrator is suboptimal, and increase the weight of reward functions for which our current prediction is suboptimal. However, the amount of increase or decrease is now related to the degree of suboptimality (rather than being binary as before) and is controlled by the step-size hyperparameter $\gamma$ (instead of doubling or completely zeroing the weight).

With the update in Eq. (8), the total weight in the system, $W_t$, remains monotone non-increasing. Moreover, by keeping track of the weights $w^{(t)}(r)$, we can conclude the following (see Appendix E.2 for derivation details and the complete proof):

**Theorem 4** (Online Regret). *For any finite $\mathcal{R}$ and any sequence $((x_t, y_t))_{t \in \mathbb{N}}$, after any number of rounds $T$ of Algorithm 1 with any $0 < \gamma < 1$, we have:*

$$\text{for all } r \in \mathcal{R} \ : \quad J_r^*(T) - \widehat{J}_r(T) \ \leq \ (1 + 2\gamma)\big(J_r^*(T) - \widetilde{J}_r(T)\big) + \frac{\log|\mathcal{R}|}{\gamma}\,.$$

Theorem 3 can be seen as an edge case of Theorem 4. When the demonstrator is optimal, as in Theorem 3, we have $J_r^*(T) - \widetilde{J}_r(T) = 0$ and we can take $\gamma$ to be an arbitrary constant (e.g.,

---

**Algorithm 2** Statistical Learner via Online-to-Batch Conversion

---

**Input:** Samples $S = \{(x_i, y_i) : i \in [m]\}$, and an online learning algorithm $\mathcal{A}_{\text{online}}$.

- Run once over $S$ an online learning algorithm $\mathcal{A}_{\text{online}}$. Record the policies $\{\widehat{\pi}_t\}_{t \in [m]}$ used at different rounds.
- For every test context $x \in \mathcal{X}$, pick a stopping time $I \sim \text{Unif}(\{1, \ldots, m\})$, and output according to $\widehat{\pi}_I(\cdot \mid x)$. In other words, output a randomized mixed policy:

$$\widehat{\pi}_{\text{o2b}}(\cdot \mid x) = \frac{1}{m} \sum_{t=1}^{m} \widehat{\pi}_t(\cdot \mid x). \tag{10}$$

---

$\gamma = 1$ in the specialized analysis of Theorem 3). When the demonstrator is suboptimal, we need to choose $\gamma$ to balance the two terms on the right-hand side, and doing so yields a regret bound of $O\left(\sqrt{\left(J_r^*(T) - \widetilde{J}_r(T)\right) \cdot \log |\mathcal{R}|}\right) = O(\sqrt{T \log |\mathcal{R}|})$. Since our final goal is a statistical guarantee as in Definition 1, it will be easier to set $\gamma$ after an online-to-batch conversion, which we do in the next subsection.

## 4.4 ANALYZING STATISTICAL ONLINE-TO-BATCH ESTIMATOR

We now return to the statistical setting of Definition 1 and consider learning a (randomized) policy given a training set $S \sim (\mathcal{D} \times \widetilde{\pi})^m$. We do so using a standard online-to-batch conversion of Algorithm 1, as specified in Algorithm 2: we run Algorithm 1 on the training set $S$ (we do not need to randomize over the ordering), keeping track of the $m$ policies $\widehat{\pi}_t$ used at each step $t = 1, \ldots, m$. The output is then a stochastic policy $\widehat{\pi}_{\text{o2b}}$, where for any context $x$, $\widehat{\pi}_{\text{o2b}}(\cdot \mid x)$ is uniform over the $m$ answers $\{\widehat{\pi}_t(x)\}_{t=1,\ldots,m}$ that the algorithm would have given for context $x$ at different steps. Equivalently, given a context $x$, $\widehat{\pi}_{\text{o2b}}$ picks a random stopping time $t \in \{1, \ldots, m\}$, runs Algorithm 1 on the first $t$ examples, and then uses the policy $\widehat{\pi}_t$ reached after $t$ steps by outputting $\widehat{\pi}_t(x)$.

To obtain an "optimistic" rate that interpolates between $1/\varepsilon$ sample complexity for optimal demonstrators and $1/\varepsilon^2$ in the worst case, we also rely on a bound $\Delta$ on the suboptimality of the demonstrator, which we can always take as $\Delta = 1$ to allow any demonstrator (since even the worst demonstrator is 1-suboptimal). An online-to-batch analysis (see details and proof in Section E) then ensures:

**Theorem 5** (Statistical Guarantee via Online-to-Batch Conversion). *Consider any finite reward class $\mathcal{R}$ containing bounded reward functions.*
*(i) For any sub-optimality bound $\Delta \in [0, 1]$, consider running Algorithm 2 with*

$$\gamma = \min\left(\tfrac{1}{2}, \sqrt{\frac{2 \log |\mathcal{R}| + 6 \log(2\delta^{-1})}{6m\Delta}}\right).$$

*For any $\delta \in (0, 1)$, sample size $m$, and demonstrator $\widetilde{\pi}$, with probability at least $1 - \delta$ over $S \sim (\mathcal{D} \times \widetilde{\pi})^m$, for all $r \in \mathcal{R}$ s.t. $\widetilde{\pi}$ has suboptimality $V_r^* - V_r(\widetilde{\pi}) \leq \Delta$, we have:*

$$V_r(\widetilde{\pi}) - V_r(\widehat{\pi}_{\text{o2b}}) \leq 2\sqrt{\frac{6\,\Delta\left(8 \log |\mathcal{R}| + 6 \log\left(\frac{2}{\delta}\right)\right)}{m}} + \frac{16 \log |\mathcal{R}| + 12 \log\left(\frac{2}{\delta}\right)}{m}. \tag{11}$$

*(ii) **Optimal Demonstrations:** Consider running Algorithm 2 with $\gamma = 1/2$. For any $\delta \in (0, 1)$ and sample size $m$, with probability at least $1 - \delta$ over $x_1, \ldots, x_m \sim_{iid} \mathcal{D}$, and any $y_1, \ldots, y_m$, for any $r \in \mathcal{R}$ for which $\forall_{i \in [m]} y_i \in \sigma_r(x_i)$, we have:*

$$V_r^* - V_r(\widehat{\pi}_{\text{o2b}}) \leq \frac{16 \log |\mathcal{R}| + 12 \log\left(\frac{2}{\delta}\right)}{m}. \tag{12}$$

See Appendix E.3 for the proof. Theorem 5 ensures that any finite reward class is learnable from demonstrations (Definition 1) with sample complexity $O(\varepsilon^{-2}(\log |\mathcal{R}| + \log \delta^{-1}))$, by always choosing $\Delta = 1$ as an upper bound on the demonstrator's suboptimality (in which case the guarantee in (11) holds for all $r \in \mathcal{R}$). Moreover, reward classes are learnable from optimal demonstrations with

sample complexity $O(\varepsilon^{-1}(\log|\mathcal{R}| + \log\delta^{-1}))$, by choosing $\Delta = 0$ and thus $\gamma = 1/2$. In fact, the theorem delivers more than what is required under Definition 1, since the optimal demonstrations $y_i$ may be chosen as arbitrary optimal actions given the contexts $x_i$, even adaptively, without following a fixed optimal policy. The guarantee holds as long as the contexts are drawn independently from $\mathcal{D}$.

**Remark 2.** We emphasize that Algorithm 1 guarantees low value suboptimality with respect to $r_\star$ by establishing guarantees uniformly over the entire reward class $\mathcal{R}$ (see Theorems 3 to 5). The learner only has access to demonstrations. It achieves a small regret/value gap by ensuring its performance is good simultaneously for all reward functions in $\mathcal{R}$, and in particular for whatever may be the true $r_\star \in \mathcal{R}$.

**Remark 3.** In this paper we focus on contextual bandits. But our results apply more generally to MDPs with deterministic (possibly unknown) transitions. Consider an MDP model class of finite cardinality, where each model specifies a deterministic transition function and a (possibly stochastic) reward. We can reduce this to a contextual bandit setting, where the entire MDP action sequence is treated as a single contextual bandit action. For auto-regressive language generation, viewing each token as an action and the entire context as the state, the transition dynamics are fixed and known and only the reward function is unknown; the entire generated sequence of tokens can be treated as a single action $y$.

## 5 Summary and Open Issues

We study the problem of learning to answer from correct demonstrations, i.e., imitation learning or apprenticeship learning in contextual bandits. We compare and contrast learning based on the low-cardinality assumptions on the Reward Class and the Demonstrator Class. We argue that the former is strictly weaker when the demonstrations are optimal. We show that natural approaches based on likelihood maximization fail under this assumption, necessitating going beyond them. We design a learner that achieves optimal sample complexity logarithmic in the reward class size, with tight rates. Our study motivates looking beyond likelihood maximization and distribution matching, and adopt new type of methods when the harder problem of distribution matching may not be achievable.

We point to Remark 4 for some additional technical theoretical questions. We also lay out two broad directions for future research below.

**Continuous Classes.** Our analysis is for arbitrary abstract, but finite, reward classes $\mathcal{R}$, and is stated in terms of the log-cardinality $|\mathcal{R}|$. Many natural reward classes are better thought of as infinite classes with continuous parametrization, $\mathcal{R} = \{r_w : (\mathcal{X} \times \mathcal{Y}) \to \mathbb{R} \mid w \in \mathbb{R}^d\}$. Relevant choices for $\mathcal{R}$ include generalized linear (or kernel) classes with $r_w(x, y) = g(\langle w, \phi(x, y)\rangle)$ (for some specified feature map $\phi(x, y)$ and link function $g : \mathbb{R} \to \mathbb{R}$), low-rank bilinear scores, or transformer models where $r_w(x, y) \in \mathbb{R}$ is the scalar output of a transformer with weights $w$ taking the sequences $x$ and $y$ as input. Can we characterize learnability from demonstrations, and the sample complexity required, for such continuous classes as well?

**Tractable Implementation and Supervised Fine Tuning of LLMs.** Another limitation of our work is that, while the sample complexity is logarithmic in the reward class size, a naive implementation of our methods requires runtime and memory linear in the cardinality, and thus exponential in the number of parameters for the continuous classes discussed above. We envision heuristic simplifications of the algorithm that may be practically implementable. However, several research questions still remain open when it comes to the practical success of such an approach in the SFT of LLMs.

In particular, what reward class $\mathcal{R} = \{r_w\}$ should be used in practice? After pretraining, we typically only have an initialized policy $\pi_\theta$. How important is this pretraining knowledge for downstream success? How can it also be used to construct different parameterized reward classes? To what extent, and in which situations, are methods based on reward classes more successful than policy-class-based likelihood maximization? How do they affect downstream performance in other phases of posttraining? Ultimately, the key question is whether this class of methods, which performs reward hedging via iterative planning and discrimination, can serve as a genuine alternative to behavior cloning, and if so, in which regimes and for what reasons.

**Acknowledgement.** We are grateful to Yishay Mansour for bringing the work of Syed & Schapire (2007) to our attention, and to Moulin et al. for pointing us to their recent related work.

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

APPENDIX TABLE OF CONTENTS

## A  MISCELLANEOUS DEFERRED DISCUSSIONS

### A.1  LEARNING GOALS UNDER TWO ASSUMPTIONS AND A COMPARISON

In this, we formalize learning under each of the two assumptions on the Demonstrator Class and Reward Class, and argue that the latter is a strictly weaker assumption. We start by discussing some terminology.

*Demonstrator Optimality.* Although our general results do *not* require the demonstrator to be optimal, it is instructive to pay particular attention to the case where it *is* optimal. As we shall see, we can get faster convergence rates and lower sample complexity under demonstrator optimality, and this will also be important in relating between the two assumptions above in Appendix A.1. We say that the demonstrator $\widetilde{\pi}$ **acts optimally** with respect to $r_\star$ iff

$$V_{r_\star}(\widetilde{\pi}) = V_{r_\star}^* := \sup_\pi V_{r_\star}(\pi). \tag{13}$$

This optimality is equivalent to requiring that for almost surely over $x \sim \mathcal{D}$, the policy $\widetilde{\pi}(\cdot|x)$ is supported on the set $\sigma_*(x)$ of optimal actions, as in Eq. (1) (i.e., "correct answers", in the special case of binary rewards where there is always a correct answer).

Our focus is on the Reward Class Assumption and so on learning in the following sense:

**Definition 1** (Learning from Demonstrations under a **Reward Class** Assumption). *We say that a reward class $\mathcal{R}$ is learnable from demonstrations by a learning rule $\mathcal{A} : (\mathcal{X} \times \mathcal{Y})^* \to (\Delta(\mathcal{Y}))^{\mathcal{X}}$ with sample complexity $\overline{m_{\mathcal{R},\mathcal{A}}(\varepsilon, \delta)}$, if for any $\varepsilon, \delta \in (0, 1)$, and any sample size $m \geq m_{\mathcal{R},\mathcal{A}}(\varepsilon, \delta)$, for any $\mathcal{D}, \widetilde{\pi}$ and $r_\star \in \mathcal{R}$, we have that with probability at least $1-\delta$ over the training set $S \sim (\mathcal{D} \times \widetilde{\pi})^m$,*

$$V_{r_\star}(\mathcal{A}(S)) \geq V_{r_\star}(\widetilde{\pi}) - \varepsilon.$$

*Furthermore, $\mathcal{R}$ is learnable from optimal demonstrations where the same definition applies but the learner is further promised that the demonstrator $\widetilde{\pi}$ acts optimally w.r.t. $r_\star$.*

We contrast this with learnability based on a Demonstrator Class Assumption:

**Definition 2** (Learning from Demonstrations under a **Demonstrator Class** Assumption). *We say that a demonstrator policy class $\Pi$ is learnable from demonstrations by a learning rule $\mathcal{A} : (\mathcal{X} \times \mathcal{Y})^* \to (\Delta(\mathcal{Y}))^{\mathcal{X}}$ with sample complexity $\overline{m_{\Pi,\mathcal{A}}(\varepsilon, \delta)}$, if for any $\varepsilon, \delta \in (0, 1)$, and any sample size $m \geq m_{\Pi,\mathcal{A}}(\varepsilon, \delta)$, for any $\mathcal{D}$, any $\widetilde{\pi} \in \Pi$, and any unknown reward $r_\star$, we have that with probability at least $1 - \delta$ over the training set $S \sim (\mathcal{D} \times \widetilde{\pi})^m$*

$$V_{r_\star}(\mathcal{A}(S)) \geq V_{r_\star}(\widetilde{\pi}) - \varepsilon.$$

*Furthermore, $\Pi$ is learnable from optimal demonstrations where the same definition applies but the learner is further promised that the reward $r_\star$ is such that $\widetilde{\pi} \in \Pi$ is optimal w.r.t. $r_\star$.*

Let us compare the relationship between the Reward Class Assumption and the Demonstrator Class Assumption that we introduced. We will do so under demonstrator optimality, as this optimality assumption provides a link between the demonstrator and the reward function. We will argue that under demonstrator optimality, the low-cardinality Reward Class Assumption is strictly weaker than the low-cardinality Demonstrator Class Assumption.

The Reward Class Assumption, combined with the optimality of $\widetilde{\pi}$, implies that for some $r \in \mathcal{R}$, the demonstrator $\widetilde{\pi}$ is supported on the set of optimal actions $\sigma_r : \mathcal{X} \to 2^{\mathcal{Y}}$ given by:[5]

$$\sigma_r(x) = \arg \max_{y \in \mathcal{Y}} r(x, y). \tag{14}$$

Thus

$$\widetilde{\pi} \in \Pi_{\mathcal{R}} := \bigcup_{r \in \mathcal{R}} \Pi_r, \text{ where } \Pi_r := \{\pi : \mathcal{X} \to \Delta(\mathcal{Y}) \mid \forall_x \operatorname{supp} \pi(\cdot \mid x) \subseteq \sigma_r(x)\}. \tag{15}$$

So the Reward Class Assumption with $\mathcal{R}$ implies the Demonstrator Class Assumption with $\Pi_{\mathcal{R}}$. However, once $|\sigma_r(x)| > 1$, the class $\Pi_{\mathcal{R}}$ is much larger than $\mathcal{R}$. Even if $\mathcal{R}$ has finite cardinality, the class $\Pi_{\mathcal{R}}$ is continuous and infinite. Even if we discretize this probability space, $\Pi_{\mathcal{R}}$ can still be much larger, since we are introducing $(|\sigma_r(x)|-1)$ parameters *per instance* to choose the distribution over correct answers, so overall a number of parameters that scales with the domain size *and* the number of correct answers. For example, even just restricting to policies that are uniform on a subset of $\sigma_r(x)$, there are $\prod_x (2^{|\sigma_r(x)|} - 1)$ optimal policies for each $r \in \mathcal{R}$.

Under the Demonstrator Class Assumption and the optimality of the demonstrator, the class of all reward functions under which the policies in $\Pi$ are optimal can be huge as well. Nevertheless, it is sufficient to consider an artificial reward class, where rewards essentially enforce matching the support of the policies in the demonstrator class $\Pi$:

$$\mathcal{R}_{\Pi}^{0\text{-}1} := \{r_{\pi}^{0\text{-}1} : \mathcal{X} \to \{0, 1\} \mid \pi \in \Pi\} \quad \text{where} \quad r_{\pi}^{0\text{-}1}(x, y) = \mathbf{1}\{y \in \operatorname{supp} \pi(\cdot \mid x)\}. \tag{16}$$

Although the actual reward $r_\star$ may not be in $\mathcal{R}_{\Pi}^{0\text{-}1}$, if some $\widetilde{\pi} \in \Pi$ is optimal with respect to $r_\star$, then being (sub)optimal w.r.t. the corresponding $r_{\widetilde{\pi}}^{0\text{-}1}$ ensures also (sub)optimality w.r.t. $r_\star$:

---

[5]$\sigma_\star$ introduced earlier in Eq. (1) is the same as $\sigma_{r_\star}$ (i.e., $\sigma_r$ evaluated at $r = r_\star$). It is important to note that $\sigma_r(x)$ may be empty: since $\mathcal{Y}$ is infinite, the supremum $\sup_{y \in \mathcal{Y}} r(x, y)$ need not be attained. Thus, for an optimal policy to exist with respect to a given $r$, it must be the case that $\sigma_r(x)$ is non-empty almost surely $x \sim \mathcal{D}$. Our general result does not rely on the existence of an optimal policy. One may instead consider (arbitrarily large but finite) $\mathcal{Y}$, or assume that the range of $r(x, y)$ is finite (e.g., binary, or taking only 100 possible values), either of which is sufficient to guarantee the existence of an optimal policy.

**Lemma 1.** *For the demonstrator class assumption, and assuming optimality of the demonstrator, i.e., $\widetilde{\pi} \in \Pi$ is optimal w.r.t. some unknown $r_\star$, we have: (1) $\widetilde{\pi}$ is optimal w.r.t. $r_{\widetilde{\pi}}^{0\text{-}1} \in \mathcal{R}_\Pi^{0\text{-}1}$, with value $V_{r_{\widetilde{\pi}}^{0\text{-}1}}^* = 1$; and (2) for any policy $\pi$, if $\pi$ is $\varepsilon$-suboptimal to $\widetilde{\pi}$ w.r.t. $r_{\widetilde{\pi}}^{0\text{-}1}$, then $\pi$ is $\varepsilon$-suboptimal w.r.t. also $r_\star$. In other words, $V_{r_{\widetilde{\pi}}^{0\text{-}1}}(\pi) \geq 1 - \varepsilon$ implies $V_{r_\star}(\pi) \geq V_{r_\star}^* - \varepsilon$.*

See Appendix C.1 for a proof. This immediately leads to the following corollary.

**Corollary 1.** *For any demonstrator class $\Pi$, if $\mathcal{R}_\Pi^{0\text{-}1}$ is learnable from optimal demonstrations by an algorithm $\mathcal{A}$ with sample complexity $m(\varepsilon, \delta)$ (Definition 1), then $\Pi$ is learnable from optimal demonstrations by the same $\mathcal{A}$ with the same sample complexity $m(\varepsilon, \delta)$.*

Since $|\mathcal{R}_\Pi^{0\text{-}1}| \leq |\Pi|$, we see that guarantees on learning from optimal demonstrations under a low-cardinality Reward Class Assumption, imply the same guarantees on learning from optimal demonstrations under the low-cardinality Demonstrator Class Assumption. A direct corollary of Theorem 5 and Corollary 1 is that our learner continues to succeed for low-cardinality demonstrator classes, when applied to a suitable reward class.

**Corollary 2** (For low-cardinality Demonstrator Classes). *Any finite demonstrator class $\Pi$ is learnable from optimal demonstrations with sample complexity $O(\varepsilon^{-1}(\log |\Pi| + \log \delta^{-1}))$, by applying the method from Theorem 5 to the reward class $\mathcal{R}_\Pi^{0\text{-}1}$ from Eq.(16).*

From a theoretical learnability perspective, and at least under demonstrator optimality, the finiteness of reward class assumption is thus weaker, and so learning results under the Reward Class Assumptions are stronger and more general than under the Demonstrator Class Assumption.

## A.2 COMPARISON WITH SYED & SCHAPIRE (2007)

We now discuss Syed & Schapire (2007) and instantiate their results for contextual bandits. Syed & Schapire study learning in general stochastic Markov decision processes with linear reward $R^*(s) = \langle \boldsymbol{w}_*, \boldsymbol{\phi}(s) \rangle$, for $\ell_1$-norm constrained reward classes. Their setup can be mapped to contextual bandits by considering the state space $\mathcal{S} = \mathcal{X} \cup (\mathcal{X} \times \mathcal{Y})$, i.e., the states consist of all contexts as well as tuples of (context, action) pairs. While their paper requires the state and action spaces to be finite, we believe this restriction is not essential for their guarantees, since the sample complexity has no dependence on the sizes of these spaces.

For a finite class $\mathcal{R}$, we can define a feature map $\boldsymbol{\phi} : \mathcal{S} \to \mathbb{R}^d$ with $d = |\mathcal{R}|$ and express each reward as a linear predictor with unit $\ell_1$ norm. Consider an $|\mathcal{R}|$-dimensional feature map whose coordinates are indexed by $r \in \mathcal{R}$ as follows:

$$\boldsymbol{\phi}(x, y)[r] = r(x, y) \quad \text{and} \quad \boldsymbol{\phi}(x)[r] = 0.$$

Then each $r \in \mathcal{R}$ can be realized by a linear classifier $\boldsymbol{w}_r$ that is the indicator vector in the $r^{\text{th}}$ coordinate. Taking the discount factor $\gamma = 1$, their expressions for the value and feature expectation (denoted by $\mu$) (Syed & Schapire, 2007, Section 2) under this embedding become:

$$V_r(\pi) = \mathbb{E}_{(x,y) \sim \mathcal{D} \times \pi} \left[ \sum_{h=0}^1 \langle \boldsymbol{w}_r, \boldsymbol{\phi}(s_h) \rangle \right] = \mathbb{E}_{(x,y) \sim \mathcal{D} \times \pi}[\langle \boldsymbol{w}_r, \boldsymbol{\phi}(x, y) \rangle] = \mathbb{E}_{(x,y) \sim \mathcal{D} \times \pi}[r(x, y)]$$

$$= \mathbb{E}_{(x,y) \sim \mathcal{D} \times \pi}[\boldsymbol{\phi}(x, y)[r]] = \mu(\pi)[r]. \tag{17}$$

Under this correspondence, we believe their Theorem 2 can be adapted to show that any finite class $\mathcal{R}$ is learnable with sample complexity $O\left(\varepsilon^{-2}(\log |\mathcal{R}| + \log \delta^{-1})\right)$.[6] Their Algorithm 1 is similar to our strategy in that it maintains weights over the features, which correspond to weights over

---

[6]We note that their Theorem 2 is stated for a multi-step MDP with discounted rewards. One needs to adapt their proof to obtain the single-step guarantee. Namely, it introduces $\epsilon_H$ as the error term incurred by a trajectory of finite length. But in contextual bandits, $\epsilon_H = 0$ even for a length-one trajectory, and thus the condition required on $H$ is degenerate. Also, the dependence on $1/(1 - \gamma)$ in the sample complexity is due to the scale of the cumulative reward, but even when taking $\gamma = 1$ in contextual bandits, the total reward is bounded in the single-step case. Thus, we can still obtain a good estimate of the value of a policy with a small $\epsilon_F$ using $O(\log(|\mathcal{R}|/\delta)/\epsilon_F^2)$ samples. Finally, since we can exactly embed $\mathcal{R}$ as a linear threshold, we have $\epsilon_R = 0$, and assuming we can solve $\arg \max$ (as we do), even $\epsilon_P = 0$.

| Our Learner | Syed & Schapire (2007) | Moulin et al. (2025) |
|:---:|:---:|:---:|
| $\sqrt{\dfrac{\Delta \log(|\mathcal{R}|/\delta)}{m}} + \dfrac{\log(|\mathcal{R}|/\delta)}{m}$ | $\sqrt{\dfrac{\log(|\mathcal{R}|/\delta)}{m}}$ | $\tilde{O}\left(\sqrt[4]{\dfrac{\log|\mathcal{R}|\log|\mathcal{Y}|}{m}}\right)$ |

Table 2: Comparison between rates of our learner and relevant prior work. For value suboptimality based on $m$ samples, our learner has a decay rate that interpolates between $1/m$ (optimal demonstrator where $\Delta = 0$) and $1/\sqrt{m}$ (always), based on the suboptimality gap $\Delta$.

rewards in our setting. The algorithm then (i) produces an optimal policy according to the reward given by the weighted combination of features (as in Line 2 of our Algorithm 1), followed by (ii) a multiplicative weight update over features (corresponding to the reward updates in Line 4 of our algorithm).

The main differences are as follows. To perform the multiplicative-weights update, their algorithm estimates the value of the current policy with respect to *all* rewards in the class to accuracy $\epsilon_F$, and then updates based on the resulting value differences (equivalently, feature-expectation differences via (17)). This leads to a multi-pass batch algorithm. Our algorithm, in contrast, is a simpler one-pass online approach: we update the weights after processing each example, without revisiting the data.

We obtain faster optimistic rates, interpolating between $O(1/\varepsilon)$ when the demonstrator is optimal and $O(1/\varepsilon^2)$ in general; see Table 2. Our learner also enjoys regret minimization (Theorem 4) or a mistake bound (Theorem 3) for adversarial sequences of demonstrations in the online setup (Figure 1). Even in the statistical setting, our learner continues to enjoy these guarantees when the demonstrations are arbitrarily adaptive, so long as they are good (Theorem 5).

**Dual Updates.** Another relevant work is Moulin et al. (2025), which can also be instantiated to admit a guarantee for contextual bandits. They again study general MDPs under a certain type of realizability assumption on the $Q$-function. In the contextual bandit case, the reward class $\mathcal{R}$ and their $Q$-function class $\mathcal{Q}$ are equivalent, and their assumption simplifies to assuming only a reward class, as in our setting. In contrast to our approach (and Syed & Schapire (2007)), which makes multiplicative weight updates over rewards, their algorithm makes iterative multiplicative updates over the space of policies, which can be viewed as dual updates. By adapting (Moulin et al., 2025, Theorem 2, Algorithm 2), we believe it is possible to show that their learner achieves $\varepsilon$-value-suboptimality with $O\left(\varepsilon^{-4}\log|\mathcal{Y}|\log(|\mathcal{R}|/\varepsilon)\right)$ samples. It is unclear whether the dependence on $\log|\mathcal{Y}|$ can be avoided for the dual update strategy. Note that $\log|\mathcal{Y}|$ may scale with the length of the responses in the autoregressive generation setting, and thus, at least from a purely statistical point of view, may not be desirable.

### A.3 REWARD MAXIMIZATION VS DISTRIBUTION MATCHING

In this paper, we view *reward maximization* or *utility* as our goal when learning from demonstrations. That is, we are measuring success not by how well we match the actions of the demonstrator policy, but only by how well we match its reward or value. This is in line with other work on imitation learning (e.g., Pomerleau, 1988; Ng et al., 2000; Syed & Schapire, 2007; Rajaraman et al., 2020; Rashidinejad et al., 2021; Foster et al., 2024, etc.), which also view the demonstrator as a resource, but the ultimate goal is to obtain good reward.

**Cloning.** One approach to obtaining good reward based on demonstrations is to match the action distribution, or *clone*, the demonstrator policy, as we have seen in Proposition 2. Indeed, the approach of Foster et al. ensures high reward by virtue of cloning the demonstrator, relying on a low-cardinality Demonstrator Class Assumption.

An important message we emphasize is that matching the distribution of the demonstrator is *not* necessary for matching the demonstrator's reward. In fact, matching the distribution (i.e. "cloning") might not even be possible, even if it is possible to match the reward.

Specifically, we show that, under a Reward Class Assumption, it is possible to match the reward of the demonstrator, but not through cloning or a maximum likelihood approach. To emphasize that

cloning is not possible in this setting, it is sufficient to consider trivial reward classes where reward maximization is straightforward, but for which cloning the demonstrator is impossible.

**Observation 1** (Informal, formalized in Proposition 3). *Consider a simple reward class $\mathcal{R} = \{r_\star\}$, containing a single reward function $r_\star(x, y) = 1$ which considers all answers to all questions as correct, with multiple answers $|\mathcal{Y}| > 1$ and infinitely many questions $\mathcal{X}$. Then reward maximization is possible without any samples, but matching the distribution of a specific optimal demonstrator $\widetilde{\pi}$ (under TV, Hellinger, KL, or reverse KL) is impossible using any number of samples.*

As such, any method (such as ours) that is guaranteed to succeed under the Reward Class Assumption must do so not by ensuring cloning. Furthermore, departing from the Demonstrator Class Assumption and cloning also entails looking beyond maximum likelihood estimation (i.e., log-loss minimization). The likelihood maximization is inherently associated with, and typically analyzed through, cloning. Indeed, in Section 3 we showed that, in our setting, maximum likelihood estimation fails at reward maximization—and therefore also at cloning—necessitating new approaches.

**Inverse Reinforcement Learning and Planning.** Another approach to imitation learning is inverse reinforcement learning (IRL) (Ng et al., 2000; Ziebart et al., 2008), which aims to recover an underlying reward function that rationalizes the expert's behavior.[7] If we have such a reward $\widehat{r}$, then one can plan a good policy for apprenticeship learning. Again this goal is impossible to achieve just based on demonstrations, and is a more difficult task than just learning a policy (i.e., apprenticeship learning).

**Reward Hedging via Iterative Discriminating and Planning.** Importantly, learning a policy from demonstrations still remains possible under the weaker Reward Class Assumption. To do so, all methods—including ours, Syed & Schapire (2007), and Moulin et al. (2025)—rely on *iterative reward hedging* and *planning*, interleaving between: (i) finding a reward that best discriminates the demonstrator from the current policy, and (ii) arriving at a new policy based on this reward. Crucially, we emphasize that this reward is *not* the true reward, nor even a reward that rationalizes the demonstrator. We also note that this iterative hedging approach is also seen and referred to as apprenticeship learning via IRL in the seminal work of Abbeel & Ng (2004). We explicitly distinguish this from IRL, since the functional role of these rewards is to discriminate a currently maintained policy from the demonstrator policy, rather than to rationalize the demonstrations—the original goal of IRL (Ng et al., 2000).

| Approaches to Apprenticeship Learning | Possibility (?) for small Reward Classes | Some Relevant References |
|---|---|---|
| Cloning | ✗ | Foster et al. (2024) |
| IRL | ✗ | Ng et al. (2000); Ziebart et al. (2008) |
| Reward Hedging via Iterative Discriminating & Planning | ✓ | Abbeel & Ng (2004); Syed & Schapire (2007); Swamy et al. (2021); Moulin et al. (2025); Ours |

**Question-Answering with Language Models.** For question-answering or prompt-completion with LLMs, the emphasis on reward maximization translates to a focus on producing good answers (i.e., correct or high-utility ones). We do not view matching the distribution of answers as a goal. This aligns with how LLM-based AI agents are used and evaluated on benchmarks (OpenAI, 2023; DeepSeekAI, 2025; Anthropic, 2024; DeepMind, 2023). Reward maximization is also the view taken in post-training of LLMs. Even in limited cases where the answer distribution might matter—such as when there is no clear consensus—the model should arguably produce a more comprehensive response that reflects its uncertainty (e.g., Kirchhof et al., 2025), rather than attempt to match the distribution of some specific training "population".

Taking reward maximization as the goal for LLMs, the next important question is: what methods should we adopt when learning from demonstrations available during SFT? The predominant ap-

---

[7]We use "rationalize" informally. While defining IRL requires care and can be ill-posed (e.g., see discussion in Ziebart et al., 2008), we believe one can adopt a notion under which the discussion remains meaningful. For example, one can aim to recover a reward function that correctly classifies the demonstrator's action on future examples drawn from a distribution that, with probability $1/2$ each, either shows the demonstrator's actual action (guaranteed to be good) or an unrelated action (not good under the ground-truth reward).

proach is log-loss minimization (i.e., Maximum Likelihood), which as discussed earlier, succeeds through distribution matching under the small Demonstrator Class Assumption. Since supervised fine-tuned models are not always satisfactory (e.g., Ji et al., 2023; OpenAI, 2023), it may be wise to question this assumption and approach. Perhaps in these SFT problems, distribution matching is impossible (with the policy class used to model the demonstrator and the amount of data).

Likelihood maximization/log-loss minimization implicitly encourages distribution matching which is a more difficult task. Kalai & Vempala (2024) and Kalai et al. (2025) go even further and argue that distribution matching and reward maximization (i.e., accuracy) might even be in tension with each other, and that insisting on distribution matching could hurt the reward. As discussed, even in situations where matching the distribution could be important, the model should indicate its uncertainty. Thus the focus should perhaps be more directly and even exclusively on *utility* or *reward*, rather than attempting to solve the more difficult task of matching the distribution—a sentiment well-captured in a quote by Vladimir Vapnik:

> *"When solving a problem of interest,*
> *do not solve a more general problem as an intermediate step."*

It might also be impossible to recover a good reward model only from the demonstrations (like in the IRL goal), without other types of feedback. Despite this, learning a good policy by adopting iterative reward hedging and planning type methods might be successful.

## B  TECHNICAL PRELIMINARY LEMMAS

We start by a technical lemma about the one-sided change of measure bound on an expectation of a bounded function in terms of the Hellinger distance (e.g., Lemma A.11 from Foster et al., 2021). We will use the exact variant from (Foster et al., 2024, Lemma 3.11).

**Lemma 2** (Change-of-measure bound via Hellinger distance, Foster et al. (2024)). *Let $(\mathcal{Z}, \mathcal{F})$ be a measurable space and let $\mathbb{P}, \mathbb{Q}$ be probability measures on it. For every measurable function $h : \mathcal{Z} \to \mathbb{R}$:*

$$\left| \mathbb{E}_{\mathbb{P}}[h] - \mathbb{E}_{\mathbb{Q}}[h] \right| \ \leq \ \sqrt{\frac{\mathbb{E}_{\mathbb{P}}[h^2] + \mathbb{E}_{\mathbb{Q}}[h^2]}{2}} \ D_{\mathsf{H}}(\mathbb{P}, \mathbb{Q}). \tag{18}$$

*In particular for $h : \mathcal{Z} \to [0, R]$,*

$$\mathbb{E}_{\mathbb{P}}[h] \ \leq \ 2\,\mathbb{E}_{\mathbb{Q}}[h] \ + \ R\,D_{\mathsf{H}}^2(\mathbb{P}, \mathbb{Q}) \,. \tag{19}$$

We now specify Freedman's inequality that allows for a tight non-asymptotic control on the martingale difference sequence. We will use Freedman's inequality for the analysis, the variant (Foster & Rakhlin, 2023, Lemma 35) in particular.

**Lemma 3** (Freedman's inequality; Bernstein for martingales). *Let $(X_t)_{t \leq T}$ be a real-valued martingale difference sequence adapted to a filtration $(\mathcal{F}_t)_{t \leq T}$. If $|X_t| \leq R$ almost surely, then for any $\eta \in (0, 1/R)$, with probability at least $1 - \delta$, for all $T' \leq T$,*

$$\sum_{t=1}^{T'} X_t \ \leq \ \eta \sum_{t=1}^{T'} \mathbb{E}_{t-1}[X_t^2] \ + \ \frac{\log(\delta^{-1})}{\eta}.$$

**Maximum Likelihood Estimation for Distribution Learning.**  We now state guarantee for the maximum likelihood estimator (MLE) for density estimation, exactly similar to (Foster et al., 2024, Section B.4). Given a class of candidate densities $\mathcal{G}$ and i.i.d. samples $z_1, \ldots, z_m \sim g_*$ (possibly not in $\mathcal{G}$), we define the empirical negative log-likelihood (log-loss) of $g \in \mathcal{G}$ as

$$L_{\log}(g) \ = \ -\sum_{i=1}^{m} \log g(z_i).$$

The maximum likelihood estimator is then

$$\hat{g}_{\mathrm{mle}} \ \in \ \arg\min_{g \in \mathcal{G}} \ L_{\log}(g). \tag{20}$$

**Definition 3** (Log-loss covering number). *For a class $\mathcal{G} \subseteq \Delta(\mathcal{Z})$, we say that a subset $\mathcal{G}' \subseteq \Delta(\mathcal{Z})$ is an $\varepsilon$-cover with respect to the log-loss if for all $g \in \mathcal{G}$ there exists $g' \in \mathcal{G}'$ such that $\sup_{z \in \mathcal{Z}} \log(g(z)/g'(z)) \leq \varepsilon$. We denote the size of the smallest such cover by $\mathcal{N}_{\log}(\mathcal{G}, \varepsilon)$.*

We have the following property of MLE's convergence in the squared Hellinger distance with high probability.

**Proposition 2.** *With probability $1 - \delta$ over $m$ i.i.d. samples from any $g_* \in \mathcal{G}$,*

$$D_{\mathsf{H}}^2(g_*, \hat{g}_{\mathrm{mle}}) \leq \inf_{\varepsilon > 0} \left\{ \frac{6 \log\big(2\,\mathcal{N}_{\log}(\mathcal{G}, \varepsilon)/\delta\big)}{m} + 4\varepsilon \right\} + 2 \inf_{g \in \mathcal{G}} \log\big(1 + D_{\chi^2}(g_* \,\|\, g)\big) + 2\,\varepsilon_{\mathrm{opt}}.$$

*In particular, if $\mathcal{G}$ is finite and $\varepsilon_{opt} = 0$, the maximum likelihood estimator satisfies*

$$D_{\mathsf{H}}^2(g_*, \hat{g}_{\mathrm{mle}}) \leq \frac{6 \log\big(2\,|\mathcal{G}|/\delta\big)}{m} + 2 \inf_{g \in \mathcal{G}} \log\big(1 + D_{\chi^2}(g_* \,\|\, g)\big).$$

*Note that the term $\inf_{g \in \mathcal{G}} \log\big(1 + D_{\chi^2}(g_* \,\|\, g)\big)$ corresponds to the misspecification error, and is zero if $g_* \in \mathcal{G}$.*

We note that the proof of (Foster et al., 2024, Proposition B.1) contains a couple of minor typographical errors, noted here for posterity. Namely, in Eq.(20) therein, the authors aim to compare $\widetilde{g}$ and $\widehat{g}$, but ended up comparing $\widetilde{g}$ and $g_*$. A similar mistake is repeated a couple of more times later in the proof without affecting the correctness of the argument.

## C  Proofs of Simple Lemmas

### C.1  Proof of Lemma 1 and Corollary 1

*Proof of Lemma 1.* Recall that $r_{\widetilde{\pi}}^{0\text{-}1}$ is the 0-1 reward that assigns reward 1 exactly when the action lies in the support of $\widetilde{\pi}(\cdot \mid x)$, and 0 otherwise. Hence, by construction, $\widetilde{\pi}$ achieves reward 1 almost surely under $r_{\widetilde{\pi}}^{0\text{-}1}$, which implies $V_{r_{\widetilde{\pi}}^{0\text{-}1}}(\widetilde{\pi}) = 1$, which is maximum possible and thus, $V_{r_{\widetilde{\pi}}^{0\text{-}1}}^* = 1$. So $\widetilde{\pi}$ is optimal w.r.t. $r_{\widetilde{\pi}}^{0\text{-}1} \in \mathcal{R}_{\Pi}^{0\text{-}1}$.

For the second part, suppose $V_{r_{\widetilde{\pi}}^{0\text{-}1}}(\pi) \geq 1 - \varepsilon$. By the definition of the 0-1 reward $r_{\widetilde{\pi}}^{0\text{-}1}$, this means that under $x \sim \mathcal{D}$ and $y \sim \pi(\cdot \mid x)$, with probability at least $1 - \varepsilon$ over $x \sim \mathcal{D}$, we have $y \in \mathrm{supp}(\widetilde{\pi}(\cdot \mid x))$. Since $\widetilde{\pi}$ is optimal w.r.t. $r_\star$, the optimal action set $\mathrm{supp}(\widetilde{\pi}(\cdot \mid x)) \subseteq \sigma_\star(x)$ almost surely over $x \sim \mathcal{D}$. Summarizing this argument below where $(x, y) \sim \mathcal{D} \times \pi$:

$$\mathbb{P}_{(x,y) \sim \mathcal{D} \times \pi}(y \in \sigma_\star(x)) = \mathbb{E}[\mathbb{1}\{y \in \sigma_\star(x)\}] \geq \mathbb{E}[\mathbb{1}\{y \in \mathrm{supp}(\widetilde{\pi}(\cdot \mid x))\}] = V_{r_{\widetilde{\pi}}^{0\text{-}1}}(\widetilde{\pi}) \geq 1 - \varepsilon.$$

Thus,

$$\mathbb{P}_{(x,y) \sim \mathcal{D} \times \pi}(y \notin \sigma_\star(x)) \leq \varepsilon. \tag{21}$$

This immediately implies the desired claim implies the desired claim according to the following calculation, which $(x, y) \sim \mathcal{D} \times \pi$:

$$\begin{aligned}
V_{r_\star}(\pi) &= \mathbb{E}_{(x,y) \sim \mathcal{D} \times \pi}[r_\star(x, y)] \\
&\geq \mathbb{E}_{(x,y) \sim \mathcal{D} \times \pi}\left[\mathbb{1}\{y \in \sigma_\star(x)\} r_\star(x, y)\right] \\
&= \mathbb{E}_{(x,y) \sim \mathcal{D} \times \pi}\left[\mathbb{1}\{y \in \sigma_\star(x)\} \sup_{y' \in \mathcal{Y}} r_\star(x, y')\right] \\
&= \mathbb{E}_{x \sim \mathcal{D}}\left[\sup_{y' \in \mathcal{Y}} r_\star(x, y') \mathbb{E}_{y \sim \widetilde{\pi}(\cdot|x)}\left[\mathbb{1}\{y \in \sigma_\star(x)\}\right]\right] \\
&= \mathbb{E}_{x \sim \mathcal{D}}\left[\sup_{y' \in \mathcal{Y}} r_\star(x, y')\left(1 - \mathbb{P}_{y \sim \pi(\cdot|x)}(y \notin \sigma_\star(x))\right)\right] \\
&\geq \mathbb{E}_{x \sim \mathcal{D}}\left[\sup_{y' \in \mathcal{Y}} r_\star(x, y')\right] - \mathbb{E}_{x \sim \mathcal{D}}\left[\mathbb{P}_{y \sim \pi(\cdot|x)}(y \notin \sigma_\star(x))\right] && \text{(since } r_\star(x, y') \leq 1\text{)} \\
&\geq \mathbb{E}_{x \sim \mathcal{D}}\left[\sup_{y' \in \mathcal{Y}} r_\star(x, y')\right] - \varepsilon && \text{(By Eq. 21)} \\
&= V_{r_\star}^* - \varepsilon.
\end{aligned}$$

$\square$

*Proof of Corollary 1.* This follows directly from Lemma 1. Suppose we can learn the reward class $\mathcal{R}_\Pi^{0\text{-}1}$ with sample complexity $m(\varepsilon, \delta)$ (cf. Definition 1). Then for any demonstrator $\widetilde{\pi} \in \Pi$ and any reward function $r_\star$ for which $\widetilde{\pi}$ is optimal, with probability at least $1 - \delta$ over the draw of $m(\varepsilon, \delta)$ samples, the output policy $\widehat{\pi}$ satisfies

$$V_{r_\star}(\widehat{\pi}) \geq V_{r_\star}^* - \varepsilon.$$

Here we used Lemma 1–$\widetilde{\pi}$ is optimal for $r_{\widetilde{\pi}}^{0\text{-}1} \in \mathcal{R}_\Pi^{0\text{-}1}$ with $V_{r_{\widetilde{\pi}}^{0\text{-}1}}^* = 1$, and any policy that is $\varepsilon$-suboptimal under $r_{\widetilde{\pi}}^{0\text{-}1}$ is also $\varepsilon$-suboptimal under $r_\star$. Therefore, learning $\mathcal{R}_\Pi^{0\text{-}1}$ with $m(\varepsilon, \delta)$ samples immediately yields a learner for $\Pi$ from optimal demonstrations (cf. Definition 2) with the same sample complexity $m(\varepsilon, \delta)$. $\square$

### C.2 IMPOSSIBILITY OF CLONING: FORMALIZING OBSERVATION 1

We formalize Observation 1 in the following proposition.

**Proposition 3.** *There exists a simple 0-1 reward class $\mathcal{R} = \{r_\star\}$ with $|\mathcal{R}| = 1$, $\mathcal{X} = \mathbb{N}$ and $\mathcal{Y} = \{0, 1\}$, so that it is trivial to achieve $V_{r_\star}(\widehat{\pi}) = 1$ with zero samples. However, for any sample size $m \in \mathbb{N}$ there exists an input distribution $\mathcal{D}$ over $\Delta(\mathcal{X})$ such that distribution matching of some optimal demonstrator (with value 1) is impossible in the following sense:*

$$\inf_{\widehat{\pi}} \sup_{\widetilde{\pi} \in \Pi_{r_\star}} \mathbb{E}_{S \sim (\mathcal{D} \times \widetilde{\pi})^m} [\mathbb{E}_{x \sim \mathcal{D}} D_{\mathsf{H}}(\widetilde{\pi}(\cdot \mid x), \widehat{\pi}(\cdot \mid x; S))]$$

$$\geq \inf_{\widehat{\pi}} \sup_{\widetilde{\pi} \in \Pi_{r_\star}} \mathbb{E}_{S \sim (\mathcal{D} \times \widetilde{\pi})^m} [\mathbb{E}_{x \sim \mathcal{D}} D_{\mathsf{TV}}(\widetilde{\pi}(\cdot \mid x), \widehat{\pi}(\cdot \mid x; S))] \geq \frac{1}{4},$$

$$\inf_{\widehat{\pi}} \sup_{\widetilde{\pi} \in \Pi_{r_\star}} \mathbb{E}_{S \sim (\mathcal{D} \times \widetilde{\pi})^m} [\mathbb{E}_{x \sim \mathcal{D}} D_{\mathsf{KL}}(\widetilde{\pi}(\cdot \mid x) \| \widehat{\pi}(\cdot \mid x; S))] \geq \frac{1}{4}$$

*and*

$$\inf_{\widehat{\pi}} \sup_{\widetilde{\pi} \in \Pi_{r_\star}} \mathbb{E}_{S \sim (\mathcal{D} \times \widetilde{\pi})^m} [\mathbb{E}_{x \sim \mathcal{D}} D_{\mathsf{KL}}(\widehat{\pi}(\cdot \mid x; S) \| \widetilde{\pi}(\cdot \mid x))] = \infty.$$

*Proof.* Let $\mathcal{X} = \mathbb{N}$, $\mathcal{Y} = \{0, 1\}$, and define $\forall_{x \in \mathcal{X}} \sigma_\star(x) = \{0, 1\}$. That is, for every $x \in \mathcal{X}$, both outcomes 0 and 1 are correct. This implies $V_{r_\star}(\widehat{\pi}) = 1$ can be achieved trivially with zero samples. We will now show the distribution matching to the optimal demonstration remains difficult.

Fix sample size $m \in \mathbb{N}$. Let $\mathcal{D}$ be the uniform distribution on $\{1, 2, \ldots, 2m\}$. Let $\Pi \subseteq \Pi_{r_\star}$ be the set of all deterministic policies supported on the optimal action sets $\sigma_\star(x) = \{0, 1\}$ such that for each $x$, $\widetilde{\pi}(\cdot \mid x)$ is either $\delta_0$ or $\delta_1$, chosen independently for each $x$. Thus $|\Pi| = 2^{2m}$.

By definition of Hellinger distance, $D_{\mathsf{H}}(\mathbb{P}, \mathbb{Q}) \geq D_{\mathsf{TV}}(\mathbb{P}, \mathbb{Q})$. Therefore it suffices to prove the bound for $D_{\mathsf{TV}}$. Given $m$ samples drawn from $\mathcal{D}$, at most $m$ distinct $x$ can appear. Thus

$$\mathbb{P}_{x \sim \mathcal{D}}(x \text{ unseen}) \geq \frac{1}{2}.$$

Now if we put uniform prior on the demonstrators in $\Pi$, then for any unobserved $x$ and any estimator policy $\widehat{\pi}$:

$$\mathbb{E}_{\widetilde{\pi} \sim \mathrm{Unif}(\Pi)} [D_{\mathsf{TV}}(\widetilde{\pi}(\cdot \mid x), \widehat{\pi}(\cdot \mid x; S)) \mid x \text{ unseen}] \geq \frac{1}{2}.$$

Combining these yields:

$$\mathbb{E}_{\widetilde{\pi} \sim \mathrm{Unif}(\Pi)} \mathbb{E}_S \mathbb{E}_{x \sim \mathcal{D}} [D_{\mathsf{TV}}(\widetilde{\pi}(\cdot \mid x), \widehat{\pi}(\cdot \mid x; S))] \geq \frac{1}{2} \cdot \frac{1}{2} = \frac{1}{4}.$$

Thus, by minimax principle:

$$\inf_{\widehat{\pi}} \sup_{\widetilde{\pi} \in \Pi} \mathbb{E}_S \mathbb{E}_{x \sim \mathcal{D}} [D_{\mathsf{TV}}(\widetilde{\pi}(\cdot \mid x), \widehat{\pi}(\cdot \mid x; S))]$$

$$\geq \inf_{\widehat{\pi}} \mathbb{E}_{\widetilde{\pi} \sim \mathrm{Unif}(\Pi)} \mathbb{E}_S \mathbb{E}_{x \sim \mathcal{D}} [D_{\mathsf{TV}}(\widetilde{\pi}(\cdot \mid x), \widehat{\pi}(\cdot \mid x; S))] \geq \frac{1}{4}.$$

In order to get first direction of KL, note that we have for any unseen $x$, we again have:

$$\mathbb{E}_{\widetilde{\pi} \sim \text{Unif}(\Pi)}\big[D_{\mathsf{KL}}(\widetilde{\pi}(\cdot \mid x)\|\widehat{\pi}(\cdot \mid x; S)) \mid x \text{ unseen}\,\big] \geq \frac{1}{2}.$$

Taking $\mathbb{E}_S\mathbb{E}_{x \sim \mathcal{D}}$ and applying minimax principle:

$$\inf_{\widehat{\pi}} \sup_{\widetilde{\pi} \in \Pi} \mathbb{E}_S\mathbb{E}_{x \sim \mathcal{D}}\big[D_{\mathsf{KL}}(\widetilde{\pi}(\cdot \mid x)\|\widehat{\pi}(\cdot \mid x; S))\big] \geq \frac{1}{4}.$$

For the opposite direction of KL, note that for any $\widehat{\pi}$, there exists $\widetilde{\pi} \in \Pi$ such that $\widetilde{\pi}(\cdot \mid x)$ assigns probability $0$ to a label that $\widehat{\pi}(\cdot \mid x; S)$ assigns probability non-zero on an unseen $x$, hence:

$$\mathbb{E}_{x \sim \mathcal{D}}D_{\mathsf{KL}}(\widehat{\pi}(\cdot \mid x)\|\widetilde{\pi}(\cdot \mid x)) = \infty.$$

In fact all our lower bounds hold almost surely over sampling of $S$. $\qquad\square$

# D  PROOFS FOR SECTION 3

## D.1  PROOF OF PROPOSITION 1: MLE IS GOOD FOR LOW-CARDINALITY $\Pi$

*Proof of Proposition 1.* Consider any unknown but fixed marginal distribution $\mathcal{D} \in \Delta(\mathcal{X})$. First observe that for any $S \in (\mathcal{X} \times \mathcal{Y})^*$, the joint law $\mathcal{D} \times \widehat{\pi}_{\text{mle}}$ is the MLE among all joint distributions $\{\mathcal{D} \times \pi : \pi \in \Pi\}$, because of the shared marginal distribution. Using Proposition 2, for $S \sim (\mathcal{D} \times \widetilde{\pi})^m$,

$$\mathbb{P}_S\left(D_{\mathsf{H}}^2\left(\mathcal{D} \times \widetilde{\pi}, \mathcal{D} \times \widehat{\pi}_{\text{mle}}\right) \leq \frac{6\log(2|\Pi|/\delta)}{m}\right) \geq 1 - \delta. \tag{22}$$

We use the above guarantee to obtain the different parts of the proposition.

(1) The first part directly follows by applying Lemma 2 to achieve the change of measure via the squared Hellinger distance with $h(x, y) = \sup_{y' \in \mathcal{Y}} r_\star(x, y') - r_\star(x, y)$. Clearly, $h : \mathcal{X} \times \mathcal{Y} \to [0, 1]$. Thus,

$$\mathbb{E}_{\mathcal{D} \times \widehat{\pi}_{\text{mle}}}\big[\sup_{y' \in \mathcal{Y}} r_\star(x, y') - r(x, y)\big] \leq 2\,\mathbb{E}_{\mathcal{D} \times \widetilde{\pi}}\big[\sup_{y' \in \mathcal{Y}} r(x, y') - r(x, y)\big] + D_{\mathsf{H}}^2\left(\mathcal{D} \times \widetilde{\pi}, \mathcal{D} \times \widehat{\pi}_{\text{mle}}\right)$$

$$= D_{\mathsf{H}}^2\left(\mathcal{D} \times \widetilde{\pi}, \mathcal{D} \times \widehat{\pi}_{\text{mle}}\right). \qquad \text{(since } \widetilde{\pi} \text{ is optimal)}$$

The LHS is simply $\mathbb{E}_{\mathcal{D} \times \widehat{\pi}_{\text{mle}}}[\sup_{y' \in \mathcal{Y}} r_\star(x, y') - r(x, y)] = V_{r_\star}^* - V_{r_\star}(\widehat{\pi}_{\text{mle}})$ by definition. Combined with (22), this gives the desired statement that with probability at least $1 - \delta$,

$$V_{r_\star}^* - V_{r_\star}(\widehat{\pi}_{\text{mle}}) \leq D_{\mathsf{H}}^2\left(\mathcal{D} \times \widetilde{\pi}, \mathcal{D} \times \widehat{\pi}_{\text{mle}}\right) \leq \frac{6\log(2|\Pi|/\delta)}{m}.$$

(2) The second part follows from the definition of TV and its relationship with the squared Hellinger distance: with probability at least $1 - \delta$, for any $r_\star : \mathcal{X} \times \mathcal{Y} \to [0, 1]$,

$$V_{r_\star}(\widetilde{\pi}) - V_{r_\star}(\widehat{\pi}_{\text{mle}}) \leq D_{\mathsf{TV}}\left(\mathcal{D} \times \widetilde{\pi}, \mathcal{D} \times \widehat{\pi}_{\text{mle}}\right) \leq \sqrt{D_{\mathsf{H}}^2\left(\mathcal{D} \times \widetilde{\pi}, \mathcal{D} \times \widehat{\pi}_{\text{mle}}\right)} \leq \sqrt{\frac{6\log(2|\Pi|/\delta)}{m}}.$$

The first inequality follows from the variational definition of the TV distance, the second inequality follows from the standard relationship $D_{\mathsf{TV}}(\mathbb{P}, \mathbb{Q}) \leq \sqrt{D_{\mathsf{H}}^2(\mathbb{P}, \mathbb{Q})}$, and the final inequality follows from (22) along with noticing that $\sqrt{\cdot}$ is an increasing function.

(3) For the final part, to obtain the optimistic rate, we adapt the variance-dependent decomposition of the reward suboptimality in terms of $D_{\mathsf{H}}^2(\cdot, \cdot)$ derived by Foster et al. (2024). Their Theorem 3.1 and Corollary 3.1, written for contextual bandits with rewards bounded in $[0, 1]$, give:

$$V_{r_\star}(\widetilde{\pi}) - V_{r_\star}(\widehat{\pi}_{\text{mle}}) \leq O\left(\sqrt{\frac{\sigma_{\widetilde{\pi}}^2 \log(|\Pi|\delta^{-1})}{m}} + \frac{\log(|\Pi|\delta^{-1})\log m}{m}\right).$$

Here,

$$\sigma_{\widetilde{\pi}}^2 := \mathbb{E}_{(x,y) \sim \mathcal{D} \times \widetilde{\pi}}\left[\left(\sup_{y'} r_\star(x, y') - r_\star(x, y)\right)^2\right]$$

$$\leq \mathbb{E}_{(x,y) \sim \mathcal{D} \times \widetilde{\pi}}\left[\left(\sup_{y'} r_\star(x, y') - r_\star(x, y)\right)\right] \qquad \text{(as } z^2 \leq z \text{ for } z \in [0, 1])$$

$$= V_{r_\star}^* - V_{r_\star}(\widetilde{\pi}) = \Delta_{\widetilde{\pi}}.$$

$\qquad\square$

## D.2 SIMPLE MLE FAILURES FOR LOW-CARDINALITY $\mathcal{R}$

We first show that there is a simple instance of a support class, where some MLE over the entire class $\Pi_{\mathcal{R}} := \bigcup_r \Pi_r$ fails.

*Proof of Theorem 1.* Fix any $\epsilon \in (0, 1)$. Let $\mathcal{Y} = \{0, 1\}$, $\mathcal{X} = \mathbb{N}$, and consider the reward class $\mathcal{R} = \{r_0, r_{01}\}$ defined by

$$r_0(x, y) = 1 - y \qquad \text{and} \qquad r_{01}(x, y) = 1 \quad \forall (x, y) \in \mathcal{X} \times \mathcal{Y}.$$

Thus, under $r_0$ only the response 0 is rewarded, whereas under $r_{01}$ both $\{0, 1\}$ receive reward 1. Choose the true reward $r_\star = r_0$, and let the demonstrator $\widetilde{\pi}$ be optimal under $r_\star$, meaning $\widetilde{\pi}(\cdot \mid x) = \delta_0(\cdot)$ for all $x$; hence every observed label equals 0.

Fix any sample size $m$. Let

$$q := \left\lceil \frac{m}{\epsilon} \right\rceil,$$

and let $\mathcal{D}$ be the uniform distribution on $[q] = \{1, \dots, q\}$. For a sample $S = \{(x_i, y_i)\}_{i=1}^m \sim (\mathcal{D} \times \widetilde{\pi})^m$, define the set of distinct unlabeled inputs

$$S_{\text{dis}} := \{x_i : i \in [m]\}.$$

Now consider the predictor $\widehat{\pi}_{\text{mle}}$ defined by

$$\widehat{\pi}_{\text{mle}}(\cdot \mid x) = \begin{cases} \delta_0(\cdot), & x \in S_{\text{dis}}, \\ \delta_1(\cdot), & x \notin S_{\text{dis}}. \end{cases} \tag{23}$$

We show that $\widehat{\pi}_{\text{mle}} \in \text{MLE}_{\Pi_{\mathcal{R}}}(S)$. The log-likelihood of any $\pi \in \Pi_{\mathcal{R}}$ is

$$\ell_{\log}(\pi; S) = \sum_{i=1}^m \log \pi(0 \mid x_i) = \sum_{x \in S_{\text{dis}}} N_x(S) \log \pi(0 \mid x),$$

where $N_x(S)$ counts occurrences of $x$ in $S$. This expression depends only on $\pi(0 \mid x)$ for $x \in S_{\text{dis}}$ and is maximized by setting $\pi(0 \mid x) = 1$ for all $x \in S_{\text{dis}}$. For $x \notin S_{\text{dis}}$, the likelihood imposes no constraint, and (23) provides a valid maximizer.

We now evaluate the value under the true reward. Since $r_0(x, y) = 1$ iff $y = 0$, the population value of $\widehat{\pi}_{\text{mle}}$ is

$$V_{r_0}(\widehat{\pi}_{\text{mle}}) = \mathbb{P}_{x \sim \mathcal{D}, \, \widehat{y} \sim \widehat{\pi}_{\text{mle}}(x)}[\widehat{y} = 0] = \mathbb{P}_{x \sim \mathcal{D}}(x \in S_{\text{dis}}) = \frac{|S_{\text{dis}}|}{q}.$$

Because $|S_{\text{dis}}| \leq m$ and $q \geq m/\epsilon$,

$$V_{r_0}(\widehat{\pi}_{\text{mle}}) \leq \frac{m}{q} \leq \epsilon.$$

This inequality holds deterministically for every sample $S$. Thus,

$$\mathbb{P}_{S \sim (\mathcal{D} \times \widetilde{\pi})^m}(V_{r_0}(\widehat{\pi}_{\text{mle}}) \leq \epsilon) = 1.$$

Since $\widehat{\pi}_{\text{mle}}$ is an MLE over $\Pi_{\mathcal{R}}$ and achieves value at most $\epsilon$, the theorem follows. $\qquad \square$

We now show that the attempt to restrict the capacity of the class via another natural choice of considering $\overline{\Pi}_{\mathcal{R}} = \bigcup_{r \in \mathcal{R}} \{\overline{\pi}_r\}$ also does not work, when the expert demonstrations $\widetilde{\pi}$ does not necessarily follow the distribution $\overline{\pi}_{r_\star}$, while still showing optimal demonstrations.

*Proof of Theorem 2.* Fix $\epsilon \in (0, 1)$ and set $s := \lceil 1/\epsilon \rceil$. Take $\mathcal{X} = \{x\}$ and

$$\mathcal{Y} = \{y^\star\} \cup \{a_1, \dots, a_{s-1}\} \cup \{b_1, \dots, b_s\},$$

so that $|\mathcal{Y}| = 1 + (s-1) + s = 2s = 2\lceil 1/\epsilon \rceil$, as required. We define two rewards $r_1, r_2 \in \{0, 1\}^{\mathcal{X} \times \mathcal{Y}}$ as follows:

$$r_1(x, y) = \begin{cases} 1, & y \in \{y^\star, a_1, \dots, a_{s-1}\}, \\ 0, & \text{otherwise}, \end{cases} \qquad \text{and} \qquad r_2(x, y) = \begin{cases} 1, & y \in \{y^\star, b_1, \dots, b_s\}, \\ 0, & \text{otherwise}. \end{cases}$$

Thus, at the unique context $x$, the set of $r_1$-optimal answers has size $s$, and the set of $r_2$-optimal answers has size $s + 1$, with
$$\{y : r_1(x, y) = 1\} \cap \{y : r_2(x, y) = 1\} = \{y^\star\}$$
and all other optimal answers disjoint. For each reward $r \in \mathcal{R} := \{r_1, r_2\}$ and $x \in \mathcal{X}$, define the corresponding "uniform" policy on optimal action set
$$\overline{\pi}_r(y \mid x) := \begin{cases} \frac{1}{|\sigma_r(x)|}, & y \in \sigma_r(x), \\ 0, & \text{otherwise.} \end{cases}$$

We now specify the data-generating process. Let $\mathcal{D}$ be the point mass at $x$, and choose the true reward to be $r_\star = r_2$. Let the demonstrator be the deterministic policy
$$\widetilde{\pi}(\cdot \mid x) = \delta_{y^\star}(\cdot),$$
which always outputs $y^\star$. Since $r_2(x, y^\star) = 1$, this demonstrator is optimal, i.e.,
$$V_{r_\star}(\widetilde{\pi}) = \mathbb{P}_{x \sim \mathcal{D}, \, y \sim \widetilde{\pi}(\cdot|x)} \big( r_2(x, y) = 1 \big) = 1.$$
For any sample size $m$, every dataset $S = \{(x_i, y_i)\}_{i=1}^m \sim (\mathcal{D} \times \widetilde{\pi})^m$ is almost surely equal to $\{(x, y^\star)\}^m$, i.e., $x_i = x$ and $y_i = y^\star$ for all $i$. The likelihoods of policies $\overline{\Pi}_\mathcal{R}$ are given by
$$\prod_{i=1}^m \overline{\pi}_{r_1}(y_i \mid x_i) = \Big( \frac{1}{|\sigma_{r_1}(x)|} \Big)^m = \Big( \frac{1}{s} \Big)^m, \qquad \prod_{i=1}^m \overline{\pi}_{r_2}(y_i \mid x_i) = \Big( \frac{1}{|\sigma_{r_2}(x)|} \Big)^m = \Big( \frac{1}{s+1} \Big)^m.$$
Since $\frac{1}{s} > \frac{1}{s+1}$, we have
$$\Big( \frac{1}{s} \Big)^m > \Big( \frac{1}{s+1} \Big)^m,$$
so $\overline{\pi}_{r_1}$ is the unique maximizer of the likelihood over $\overline{\Pi}_\mathcal{R}$. Therefore, for every $m$ and almost surely for $S \sim (\mathcal{D} \times \widetilde{\pi})^m$, $\overline{\pi}_{r_1}$ is the unique element of $\mathrm{MLE}_{\overline{\Pi}_\mathcal{R}}(S)$.

Finally, we evaluate the value of this MLE under the true reward $r_\star = r_2$. The policy $\overline{\pi}_{r_1}$ puts mass $1/s$ on $y^\star$ and mass $1/s$ on each of $a_1, \ldots, a_{s-1}$, and zero elsewhere. Among these, only $y^\star$ is rewarded by $r_2$. Thus
$$V_{r_\star}(\widehat{\pi}_{\mathrm{mle}}) = V_{r_2}(\overline{\pi}_{r_1}) = \mathbb{P}_{x \sim \mathcal{D}, \, \widehat{y} \sim \overline{\pi}_{r_1}(\cdot|x)} \big( r_2(x, \widehat{y}) = 1 \big) = \overline{\pi}_{r_1}(y^\star \mid x) = \frac{1}{s} = \frac{1}{\lceil 1/\epsilon \rceil} \le \epsilon.$$

Recall that the analysis holds almost surely over $S \sim (\mathcal{D} \times \widetilde{\pi})^m$, and the theorem follows. $\qquad\square$

*Proof of Theorem 2.* Fix $\gamma \in (0, 1)$ and let $s := \lceil 1/\gamma \rceil$. Take $\mathcal{X} = \{x\}$ and
$$\mathcal{Y} = \{y^\star\} \cup \{a_1, \ldots, a_{s-1}\} \cup \{b_1, \ldots, b_s\},$$
so $|\mathcal{Y}| = 1 + (s - 1) + s = 2s = 2\lceil 1/\gamma \rceil$. Define a two reward class $\mathcal{R} \subseteq \{0, 1\}^{\mathcal{X} \times \mathcal{Y}}$ such that the rewards $r_1, r_2$ have correct answer sets given by
$$\sigma_{r_1}(x) = \{y^\star, a_1, \ldots, a_{s-1}\} \quad \text{(size } s\text{)}, \qquad \sigma_{r_2}(x) = \{y^\star, b_1, \ldots, b_s\} \quad \text{(size } s+1\text{)},$$
so $\sigma_{r_1}(x) \cap \sigma_{r_2}(x) = \{y^\star\}$ and they are otherwise disjoint.
$$\overline{\Pi}_\mathcal{R} := \{\overline{\pi}_r : r \in \mathcal{R}\}, \qquad \overline{\pi}_r(y \mid x) = \begin{cases} \frac{1}{|\sigma_r(x)|}, & y \in \sigma_r(x), \\ 0, & \text{otherwise.} \end{cases}$$

Set $\mathcal{D}$ to be the point mass at $x$ and choose the ground-truth support $r_\star = r_2$ with data-generating conditional $\widetilde{\pi} = \delta_{y^\star}$ (always emit $y^\star$). For any $m$, every dataset $S \sim (\mathcal{D} \times \widetilde{\pi})^m$ equals $\{(x, y^\star)\}^m$.

It is simple to see that $\overline{\pi}_{r_1} \in \mathrm{MLE}_{\overline{\Pi}_\mathcal{R}}(S)$ is the unique maximum likelihood estimator. This is because
$$\prod_{i=1}^m \overline{\pi}_{r_1}(y_i \mid x_i) = \Big( \frac{1}{s} \Big)^m, \qquad \prod_{i=1}^m \overline{\pi}_{r_2}(y_i \mid x_i) = \Big( \frac{1}{s+1} \Big)^m.$$

However, with $r_\star = r_2$, the estimator $\widehat{\pi}_{\mathrm{mle}} = \overline{\pi}_{r_1}$ has the value
$$V_{r_\star}(\widehat{\pi}_{\mathrm{mle}}) = V_{r_\star}(\overline{\pi}_{r_1}) = \mathbb{P}_{(x, \widehat{y}) \sim \mathcal{D} \times \overline{\pi}_{r_1}} \big( \widehat{y} \in \sigma_{r_2}(x) \big) = \frac{1}{s} = \frac{1}{\lceil 1/\gamma \rceil} \le \gamma.$$

On the other other hand, $V_{r_\star}(\widetilde{\pi}) = 1$ because $y^\star \in \sigma_{r_2}(x)$. All bounds hold almost surely over the sampling of a training set $S$, hence the desired theorem holds. $\qquad\square$

### D.3 OVERLAP OF MLE

We now show that MLE over the restricted capacity class $\overline{\Pi}_{\mathcal{R}}$ attains a *non-trivial overlap* at the logarithmic sample complexity, though its failure to directly optimize the objective of interest (cf. Section 3).

**Theorem 6.** *Consider any finite class $\mathcal{R} \subseteq [0,1]^{\mathcal{X} \times \mathcal{Y}}$ and an unknown joint realizable distribution $\mathcal{D} \times \widetilde{\pi}$, where $\widetilde{\pi}$ acts optimally w.r.t. some unknown $r_\star \in \mathcal{R}$. For any estimator $\widehat{\pi}_{\mathrm{mle}} \in \mathrm{MLE}_{\overline{\Pi}_{\mathcal{R}}}(S)$, we have the following guarantee: for any sample size $m \geq \varepsilon^{-1}\left(\log|\mathcal{R}| + \log(1/\delta)\right)$ , we have*

$$\mathbb{P}_S\left(\mathbb{P}_{x \sim \mathcal{D}}\left[\exists_{y \in \sigma_\star(x)} \, \widehat{\pi}_{\mathrm{mle}}(y|x) \geq \frac{1}{\kappa}\right] \geq 1 - \varepsilon\right) \geq 1 - \delta, \quad \text{for any} \quad m \geq \tfrac{1}{\varepsilon}\left(\log|\mathcal{R}| + \log(1/\delta)\right),$$

*where $\kappa := \sup_{r,x} |\sigma_r(x)|$ are the maximum support size.*

*Proof of Theorem 6.* The proof is simple. We first describe some notation: we let

$$\sigma_r(x) = \arg\max_{y \in \mathcal{Y}} r(x,y).$$

We generalize the notion of consistency from Eq. (4) to all the reward functions that agree with the notion of optimality of the observed demonstrations in $S$:

$$\mathrm{CONS}_{\mathcal{R}}(S) = \{r \in \mathcal{R} \mid \forall_{(x,y) \in S} \; y \in \sigma_r(x)\}.$$

We start the proof by noting that the optimal policy supported on the actual ground-truth reward $r_\star$ has non-zero likelihood. Therefore, any policy in the set $\mathrm{MLE}_{\overline{\Pi}_{\mathcal{R}}}(S)$ must have non-zero likelihood as well. Thus, for any $r$ for which $\overline{\pi}_r \in \mathrm{MLE}_{\overline{\Pi}_{\mathcal{R}}}(S)$, we must have that $r \in \mathrm{CONS}_{\mathcal{R}}(S)$. The conclusion of this argument is that $\mathrm{MLE}_{\overline{\Pi}_{\mathcal{R}}}(S) \subseteq \{\overline{\pi}_r \mid r \in \mathrm{CONS}_{\mathcal{R}}(S)\}$. Therefore, in order to establish

$$\mathbb{P}_{S \sim (\mathcal{D} \times \widetilde{\pi})^m}\left(\mathbb{P}_{x \sim \mathcal{D}}\left(\exists_{y \in \sigma_\star(x)} \, \widehat{\pi}_{\mathrm{mle}}(y|x) \geq \frac{1}{\kappa}\right) \geq 1 - \varepsilon\right) \geq 1 - \delta,$$

it suffices to establish

$$\mathbb{P}_S\left(\forall r \in \mathrm{CONS}_{\mathcal{R}}(S) : \mathbb{P}_{x \sim \mathcal{D}}\left(\sigma_r(x) \cap \sigma_\star(x) \neq \emptyset\right) \geq 1 - \varepsilon\right) \geq 1 - \delta. \tag{24}$$

This is equivalent to:

$$\mathbb{P}_S\left(\forall r \in \mathrm{CONS}_{\mathcal{R}}(S) : \mathbb{P}_{x \sim \mathcal{D}}\left(\sigma_r(x) \cap \sigma_\star(x) = \emptyset\right) \leq \varepsilon\right) \geq 1 - \delta. \tag{25}$$

Consider any bad $r \in \mathcal{R}$ such that $\mathbb{P}_{x \sim \mathcal{D}}(\sigma_r(x) \cap \sigma_\star(x) = \emptyset) > \varepsilon$. With each draw $(x_i, y_i) \sim (\mathcal{D} \times \widetilde{\pi})$, we have that $r$ becomes inconsistent with probability at least $\varepsilon$, i.e., $\mathbb{P}_{(x_i, y_i) \sim (\mathcal{D} \times \widetilde{\pi})}(y_i \notin \sigma_r(x_i)) > \varepsilon$. Therefore, for any fixed $r$, over the draw of randomness of a sample $S \sim (\mathcal{D} \times \widetilde{\pi})^m$

$$\mathbb{P}_S(r \in \mathrm{CONS}_{\mathcal{R}}(S)) \leq (1 - \varepsilon)^m \leq e^{-\varepsilon m}.$$

Therefore, by a standard union bound,

$$\mathbb{P}_S\left(\exists \text{ bad } r \in \mathrm{CONS}_{\mathcal{R}}(S) \text{ that survives}\right) \leq |\mathcal{R}| \, e^{-\varepsilon m} \leq |\mathcal{R}| \, 2^{-\varepsilon m}.$$

The theorem follows by noting that the $|\mathcal{R}| \, 2^{-\varepsilon m} \leq \delta$ for any $m \geq \frac{\log|\mathcal{R}| + \log(1/\delta)}{\varepsilon}$. $\qquad\square$

**Remark 4.** It may be possible to turn this into a predictor that directly starts to produce good responses, depending on the overlap among hypotheses and other types of feedback available in post-training (e.g., whether a generated response is good or not). This overlap can be captured by a parameter that reflects the need for repeated sampling and the number of feedback that must be queried, which in turn allows for a more quantitative understanding of how many feedbacks are required to guarantee performance in terms of this parameter. However, we leave it open to formulate an interesting setup that enables a study of both types of feedbacks together for our problem, and we do not attempt to investigate this any further.

## E PROOFS FROM SECTION 4

We first start with a proof of the mistake bound and regret analysis of our Algorithm 1 for the online setup.

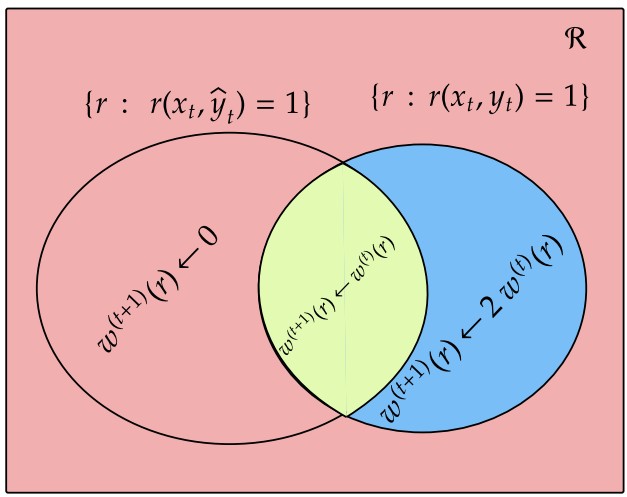

Figure 2: A visualization of the update rule of Algorithm 1 during $t^{\text{th}}$ round. The weight of the hypotheses in the red, green, and blue regions are respectively set to zero, unchanged, and doubled.

### E.1 PROOF OF THEOREM 3

*Proof of Theorem 3.* Let $W_{t+1} = \sum_{r \in \mathcal{R}} w^{(t+1)}(r)$ be the total weight in the system after completion of $t$ rounds. We first claim that the total weight $W_t$ is non-increasing, i.e., $W_{t+1} \leq W_t$. The intuition is that because $\widehat{y}_t$ is the weight-maximizer, the total weight of hypotheses such that $r(x_t, y_t) = 1$ but $r(x_t, \widehat{y}_t) \neq 1$ (i.e., the added weight) is at most the total weight of hypotheses such that $r(x_t, \widehat{y}_t) = 1$ but $r(x_t, y_t) \neq 1$ which we always zero out (see also Figure 2). Formally,

$$W_{t+1} = \sum_{r(x_t,y_t)=1 \text{ and } r(x_t,\widehat{y}_t) \neq 1} 2w^{(t)}(r) + \sum_{r(x_t,y_t)=r(x_t,\widehat{y}_t)=1} w^{(t)}(r) = \sum_{r(x_t,y_t)=1 \text{ and } r(x_t,\widehat{y}_t) \neq 1} w^{(t)}(r) + \sum_{r(x_t,y_t)=1} w^{(t)}(r)$$

$$\leq \sum_{r(x_t,y_t)=1 \text{ and } r(x_t,\widehat{y}_t) \neq 1} w^{(t)}(r) + \sum_{r(x_t,\widehat{y}_t)=1} w^{(t)}(r) = \sum_{r(x_t,y_t)=1 \text{ or } r(x_t,\widehat{y}_t)=1} w^{(t)}(r) \leq W_t, \quad (26)$$

where the equalities are rearrangements of terms inside the summations, and first inequality follows from the definition of $\widehat{y}_t$ in Algorithm 1 which implies for any $y \in \mathcal{Y}$ (so $y = y_t$ in particular):

$$\sum_{r(x_t,y)=1} w^{(t)}(r) = \sum_{r \in \mathcal{R}} w^{(t)}(r) r(x_t, y) \leq \sum_{r \in \mathcal{R}} w^{(t)}(r) r(x_t, \widehat{y}_t) = \sum_{r(x_t,\widehat{y}_t)=1} w^{(t)}(r). \quad (27)$$

Now, for any fixed $r \in \mathcal{R}$, in the mistake round for $r$—meaning $r(x_t, \widehat{y}_t) \neq 1$—the weight of $r$ is doubled. And so after any number of rounds $T$ with $M_r$ mistakes w.r.t. some consistent $r \in \mathcal{R}$:

$$w^{(T+1)}(r) = 2^{M_r} \leq W_{T+1} \leq W_1 = |\mathcal{R}|, \quad \text{which implies } M_r \leq \log |\mathcal{R}|. \qquad \square$$

### E.2 PROOF OF THEOREM 4

*Proof.* We will use the elementary inequalities

$$(1+\gamma)^u \leq 1 + \gamma u \quad \text{and} \quad (1-\gamma)^u \leq 1 - \gamma u \quad \text{for } u \in [0,1], \ \gamma \in [0,1]. \quad (28)$$

Easing the notation, let us denote

$$\lambda_t(r,y) := r(x_t, *) - r(x_t, y) = \sup_{y' \in \mathcal{Y}} r(x_t, y') - r(x_t, y).$$

We first show that the total-weight in the system $W_t = \sum_{r \in \mathcal{R}} w^{(t)}(r)$ is non-increasing.

$$
\begin{aligned}
W_{t+1} - W_t &= \sum_{r \in \mathcal{R}} w^{(t)}(r)(1+\gamma)^{\lambda_t(r,\widehat{y}_t)}(1-\gamma)^{\lambda_t(r,y_t)} - \sum_{r \in \mathcal{R}} w^{(t)}(r) \\
&\leq \sum_{r \in \mathcal{R}} w^{(t)}(r)\big(1+\gamma\lambda_t(r,\widehat{y}_t)\big)\big(1-\gamma\lambda_t(r,y_t)\big) - \sum_{r \in \mathcal{R}} w^{(t)}(r) \\
&\qquad\qquad\qquad\qquad\qquad\qquad\qquad\qquad\qquad\qquad \text{(by (28) \& range}(r) \subseteq [0,1]) \\
&= \gamma \sum_{r \in \mathcal{R}} w^{(t)}(r)\big(\lambda_t(r,\widehat{y}_t) - \lambda_t(r,y_t)\big) - \gamma^2 \sum_{r \in \mathcal{R}} w^{(t)}(r)\lambda_t(r,\widehat{y}_t)\lambda_t(r,y_t) \\
&= \gamma \sum_{r \in \mathcal{R}} w^{(t)}(r)\big(r(x_t,y_t) - r(x_t,\widehat{y}_t)\big) - \gamma^2 \sum_{r \in \mathcal{R}} w^{(t)}(r)\lambda_t(r,\widehat{y}_t)\lambda_t(r,y_t) \; \leq \; 0,
\end{aligned}
$$

where the last inequality holds because the first term is non-positive (the learner's prediction $\widehat{y}_t$ maximizes the weighted reward) and the second term is non-positive anyway. Hence $W_{t+1} \leq W_t$, and by induction $W_T \leq W_1 = |\mathcal{R}|$.

Because $\{W_t\}$ is non-increasing, and we started with $W_1 = |\mathcal{R}|$, we have that for any $r \in \mathcal{R}$

$$
W_T(r) = (1+\gamma)^{\,J_r^*(T) - \widehat{J}_r(T)}(1-\gamma)^{\,J_r^*(T) - \widetilde{J}_r(T)} \; \leq \; W_{T+1} \; \leq \; W_1 = |\mathcal{R}|.
$$

Taking logarithms and rearranging terms gives

$$
J_r^*(T) - \widehat{J}_r(T) \; \leq \; \frac{\log|\mathcal{R}|}{\log(1+\gamma)} + \frac{\log\left(\frac{1}{1-\gamma}\right)}{\log(1+\gamma)}\big(J_r^*(T) - \widetilde{J}_r(T)\big).
$$

Finally, using $\frac{1}{\log(1+\gamma)} \leq \frac{1}{\gamma}$ and $\frac{\log\left(\frac{1}{1-\gamma}\right)}{\log(1+\gamma)} \leq 1 + 2\gamma$ for $\gamma \in (0,1)$, we obtain the stated bound. $\quad\square$

We now prove Theorem 5 via an online-to-batch conversion.

### E.3   Proof of Theorem 5: Online-to-Batch Analysis via Freedman's Inequality

We now analyze the online-to-batch conversion in Algorithm 2, deriving Theorem 5 with tight optimistic rates. This is achieved by applying Freedman's inequality (Lemma 3) to an appropriate martingale difference sequence. Here we analyze Algorithm 2 only as applied to Algorithm 1, but we believe the same analysis extends to more general settings with different loss functions that admit a suitable regret guarantee, and may be of independent interest.

*Proof of Theorem 5.* We consider the natural filtration $\mathcal{F}_t$ for the probability space of the first $t$ samples, and denote by $\mathbb{E}_t[\cdot]$ expectation conditioned on this filtration.[8] Fix any $r \in \mathcal{R}$. To analyze the value of the policy, observe that

$$
V_r(\widetilde{\pi}) = \mathbb{E}_{t-1}[r(x_t,y_t)] \quad V_r(\widehat{\pi}_t) = \mathbb{E}_{t-1}[r(x_t,\widehat{y}_t)], \quad \text{and} \quad V_r^* = \mathbb{E}_{t-1}\big[\sup_y r(x_t,y)\big].
$$

Thus the following forms a martingale difference sequence:

$$
Z_t^r \; := \; \big(V_r^* - V_r(\widehat{\pi}_t)\big) - \big(\sup_y r(x_t,y) - r(x_t,\widehat{y}_t)\big)
$$

with $|Z_t^r| \leq 1$. Since $\mathbb{E}_{t-1}[Z_t^r] = 0$, the sequence $(Z_t^r)_{t=1}^m$ forms a martingale difference sequence with respect to $(\mathcal{F}_t)_{t=0}^m$. To control its conditional variance, note that

$$
\begin{aligned}
\mathbb{E}_{t-1}\big[(Z_t^r)^2\big] = \mathrm{Var}_{t-1}(Z_t^r) &= \mathrm{Var}_{t-1}\bigg(\Big[\sup_y r(x_t,y) - r(x_t,\widehat{y}_t)\Big]\bigg) \\
&\leq \mathbb{E}_{t-1}\bigg[\sup_y r(x_t,y) - r(x_t,\widehat{y}_t)\bigg] = V_r^* - V_r(\widehat{\pi}_t),
\end{aligned}
$$

---

[8]We will always assume that the quantities of interest are measurable with respect to this filtration, without delving into the formal details.

because $V_r^* - V_r(\widehat{\pi}_t)$ is $\mathcal{F}_{t-1}$-measurable and hence acts as a constant. The only inequality above follows from the fact that $\sup_y r(x_t, y) - r(x_t, \widehat{y}_t) \in [0, 1]$, and for such random variables $\mathrm{Var}[\cdot] \leq \mathbb{E}[\cdot]$.

Applying Freedman's inequality (Lemma 3) with $R = 1$ and confidence parameter $\delta/2$, for any $\eta \in (0, 1)$ we have, with probability at least $1 - \delta/2$,

$$\sum_{t=1}^m Z_t^r \;\leq\; \eta \sum_{t=1}^m (V_r^* - V_r(\widehat{\pi}_t)) + \frac{\log(2\delta^{-1})}{\eta}.$$

Expanding $Z_t^r = (V_r^* - V_r(\widehat{\pi}_t)) - (\sup_y r(x_t, y) - r(x_t, \widehat{y}_t))$ and summing over $t = 1, \ldots, m$ gives

$$\sum_{t=1}^m (V_r^* - V_r(\widehat{\pi}_t)) - \sum_{t=1}^m \left(\sup_y r(x_t, y) - r(x_t, \widehat{y}_t)\right) \;\leq\; \eta \sum_{t=1}^m (V_r^* - V_r(\widehat{\pi}_t)) + \frac{\log(2\delta^{-1})}{\eta}.$$

Dividing by $m$ and noting that the final averaged policy $\widehat{\pi}_{\mathrm{o2b}}$ is defined so that

$$V_r(\widehat{\pi}_{\mathrm{o2b}}) = \frac{1}{m} \sum_{t=1}^m V_r(\widehat{\pi}_t),$$

yields

$$(1 - \eta)(V_r^* - V_r(\widehat{\pi}_{\mathrm{o2b}})) \;\leq\; \frac{J_r^*(m) - \widehat{J}_r(m)}{m} + \frac{\log(2\delta^{-1})}{\eta m}.$$

Now setting $\eta = \gamma \in (0, 1/2]$ and using the elementary inequality $\frac{1}{1-\gamma} \leq 1 + 2\gamma$ for $\gamma \in (0, 1/2]$, we obtain

$$V_r^* - V_r(\widehat{\pi}_{\mathrm{o2b}}) \leq \frac{J_r^*(m) - \widehat{J}_r(m)}{m(1 - \gamma)} + \frac{\log(2\delta^{-1})}{\gamma(1 - \gamma)m}$$

$$\leq (1 + 2\gamma) \frac{J_r^*(m) - \widehat{J}_r(m)}{m} + \frac{2\log(2\delta^{-1})}{\gamma m}. \tag{29}$$

We now denote the suboptimality gap of the demonstrator by $\Delta_{\widetilde{\pi}}^r := V_r^* - V_r(\widetilde{\pi})$, and repeat the same analysis on the martingale difference sequence[9]

$$Z_t^r \;:=\; \left(\sup_y r(x_t, y) - r(x_t, y_t)\right) - \left(V_r^* - V_r(\widetilde{\pi})\right).$$

Applying Freedman's inequality (Lemma 3) with $R = 1$ and confidence parameter $\delta/2$, for any $\gamma = \eta \in (0, 1)$ we obtain, with probability at least $1 - \delta/2$,

$$\sum_{t=1}^m Z_t^r \leq \gamma \sum_{t=1}^m \mathbb{E}_{t-1}\left[(Z_t^r)^2\right] + \frac{\log(2\delta^{-1})}{\gamma} \leq \gamma m \Delta_{\widetilde{\pi}}^r + \frac{\log(2\delta^{-1})}{\gamma}.$$

Substituting the expression for $Z_t^r$ gives

$$\sum_{t=1}^m \left(\sup_y r(x_t, y) - r(x_t, y_t)\right) - m \Delta_{\widetilde{\pi}}^r \leq \gamma m \Delta_{\widetilde{\pi}}^r + \frac{\log(2\delta^{-1})}{\gamma}.$$

Rearranging after dividing both sides by $m$ yields

$$\frac{J_r^*(m) - \widetilde{J}_r(m)}{m} \;\leq\; (1 + \gamma) \Delta_{\widetilde{\pi}}^r + \frac{\log(2\delta^{-1})}{\gamma m}. \tag{30}$$

---

[9]For this step of the proof, however, a direct application of Bernstein's inequality also suffices, and it is not necessary to proceed via a martingale difference sequence.

Now we combine Eqs. (29) and (30) and apply a union bound over the two events. Using the regret bound from Theorem 4, we obtain the following: with probability at least $1 - \delta$,

$$V_r^* - V_r(\widehat{\pi}_{\mathrm{o2b}}) \leq \frac{(1 + 2\gamma)(J_r^*(m) - \widehat{J}_r(m))}{m} + \frac{2\log(2\delta^{-1})}{\gamma m} \tag{31}$$

$$\leq \frac{(1 + 2\gamma)^2}{m}\left(J_r^*(m) - \widetilde{J}_r(m)\right) + \frac{(1 + 2\gamma)\log|\mathcal{R}|}{\gamma m} + \frac{2\log(2\delta^{-1})}{\gamma m} \tag{32}$$

$$\leq (1 + 2\gamma)^2(1 + \gamma)\,\Delta_{\widetilde{\pi}}^r + \frac{(1 + 2\gamma)^2\log(2\delta^{-1})}{\gamma m} + \frac{(1 + 2\gamma)\log|\mathcal{R}|}{\gamma m} + \frac{2\log(2\delta^{-1})}{\gamma m} \tag{33}$$

$$\leq (1 + 6\gamma)\,\Delta_{\widetilde{\pi}}^r + \frac{2\log|\mathcal{R}|}{\gamma m} + \frac{6\log(2\delta^{-1})}{\gamma m}, \quad \text{where we used } \gamma \in (0, 1/2]. \tag{34}$$

Subtracting $\Delta_{\widetilde{\pi}}^r = V_r^* - V_r(\widetilde{\pi})$ from both sides gives, with probability at least $1 - \delta$,

$$V_r(\widetilde{\pi}) - V_r(\widehat{\pi}_{\mathrm{o2b}}) \leq 6\gamma\Delta_{\widetilde{\pi}}^r + \frac{2\log|\mathcal{R}|}{\gamma m} + \frac{6\log(2\delta^{-1})}{\gamma m}. \tag{35}$$

Finally, choosing

$$\gamma = \min\left(\tfrac{1}{2},\ \sqrt{\frac{2\log|\mathcal{R}| + 6\log(2\delta^{-1})}{6m\Delta}}\right)$$

yields for any $r \in \mathcal{R}$ for which $\Delta_{\widetilde{\pi}}^r \leq \Delta$, we have with probability at least $1 - \delta$

$$V_r(\widetilde{\pi}) - V_r(\widehat{\pi}_{\mathrm{o2b}}) \leq 2\sqrt{\frac{6\,\Delta(2\log|\mathcal{R}| + 6\log(2\delta^{-1}))}{m}} + \frac{4\log|\mathcal{R}| + 12\log(2\delta^{-1})}{m}. \tag{36}$$

Finally, applying a union bound over all $r \in \mathcal{R}$ and replacing $\delta$ with $\delta/|\mathcal{R}|$ yields the stated guarantee in Theorem 5 that holds for all $r \in \mathcal{R}$ simultaneously with probability at least $1 - \delta$, completing the proof.

The second part of the theorem follows by substituting $\Delta = 0$. In this case, the only randomness comes from the contexts $x_1, \ldots, x_m \sim_{iid} \mathcal{D}$, and we assume the relevant quantities are measurable with respect to the filtration on the probability space of the the contexts. $\square$

## F  pass@$k$ ERROR MINIMIZATION

As an extension of our setup, we also consider a pass@$k$ objective and show that, when the demonstrator is *optimal*, the minimax sample complexity is $\Theta(\log_{k+1}|\mathcal{R}|)$. We note that when the demonstrator is suboptimal, it is unclear whether one can compete with the demonstrator with an *improved* sample complexity of $O(\log_{k+1}|\mathcal{R}|)$. For a pass@$k$ metric, the policy $\mu : \mathcal{X} \to \Delta(\mathcal{Y}^k)$ is allowed to output $k$ responses $(y^{(1)}, \ldots, y^{(k)})$, and it receives the maximal reward among the candidate responses, i.e., $\max_{1 \leq i \leq k} r_\star(x, y^{(i)})$.

$$V_{r_\star}(\mu) := \mathbb{E}_{x \sim \mathcal{D}}\mathbb{E}_{\boldsymbol{y} = (y^{(1)}, \ldots, y^{(k)}) \sim \mu(\cdot|x)}\left[\max_{1 \leq i \leq k} r_\star(x, y^{(i)})\right], \tag{37}$$

For binary rewards, this corresponds exactly to pass@$k$-accuracy. Such an objective is often used in benchmarks (e.g., Chen et al. (2021); Orlanski et al. (2022); Dalal et al. (2025)) and is natural in settings where it is sensible to output multiple completions and let the user choose (e.g., in writing assistance tasks), or when verifying an answer is easier than finding one (e.g., when the answer is a Lean proof, or a Python program that can be tested, or perhaps using another LLM as a verifier). The policy $\mu$ may output any joint distribution over the $k$ actions, with arbitrary dependencies—for instance, to encourage diversity or coverage. This need not be a product distribution corresponding to repeated independent sampling from a single-response policy. Importantly, at training time, the learner still receives only a single correct answer $y_t$ for each training question $x_t$.

Our methods (Algorithm 1) and analysis can be extended to the pass@$k$ objective. In this setting, the additional flexibility available to the learner yields an improved sample complexity when the

demonstrator is optimal. The first modification is in Line 2, where now we select the $k$ actions sequentially and greedily as follows: each action $y_t^{(i)}$ is selected based on the weighted majority vote for whether $y_t^{(i)}$ is optimal w.r.t. $r$ among the yet "unsatisfied" rewards $r$ (i.e., those for which none of the preceding $i-1$ actions are good). Upon receiving $y_t$, we zero out the weights of inconsistent rewards $r$ for which the observed $y_t$ is not optimal, i.e., $y_t \notin \sigma_r(x_t)$ (as in Algorithm 1), and update

$$w^{(t+1)}(r) \leftarrow (k+1)\, w^{(t)}(r)\,, \tag{38}$$

more aggressively by a factor of $(k+1)$, for all rewards $r$ for which the optimal action set $\sigma_r(x_t)$ contains $y_t$ but none of $\{\widehat{y}_t^{(1)}, \ldots, \widehat{y}_t^{(k)}\}$. Counting a *mistake* as when *none* of the generated answer by the algorithm were optimal, this new strategy leads to a mistake bound of $\log_{k+1} |\mathcal{R}|$ in the online case (Theorem 8), which then translates to a similar sample complexity in the statistical case, using the same online-to-batch conversion (Algorithm 2).

**Theorem 7** (Learning guarantee for pass@$k$-metric)**.** *Consider any finite reward class $\mathcal{R} \subseteq [0,1]^{\mathcal{X} \times \mathcal{Y}}$. For any $k > 1$:*

- *For any sequence $((x_t, y_t))_{t \in \mathbb{N}}$, and any $r \in \mathcal{R}$ for which all the demonstrations are optimal, i.e., $\forall_t\, y_t \in \sigma_r(x_t)$, Algorithm 3 makes at most $\log_{k+1} |\mathcal{R}|$ mistake w.r.t. $r$ (i.e., $\forall_{i \in [k]}\, \widehat{y}_t^{(i)} \notin \sigma_r(x_t)$).*

- *Letting $\widehat{\mu}_{\mathrm{o2b}}$ be the policy given by Algorithm 2, for any $\delta \in (0,1)$ and sample size $m$, with probability at least $1 - \delta$, over $x_1, \ldots, x_m \sim_{iid} \mathcal{D}$, and $y_1, \ldots, y_m$ s.t. $\forall_{t \in [m]}\, y_t \in \sigma_\star(x_t)$ for some $r_\star \in \mathcal{R}$, we have:*

$$V_{r_\star}^* - V_{r_\star}(\widehat{\mu}_{\mathrm{o2b}}) \leq \frac{\log_{k+1} |\mathcal{R}| + 4\log\left(2\delta^{-1}\right)}{m}\,.$$

We provide the proof and details of the algorithm in Appendices F.1 and F.2. We show a matching lower bound of $\Omega(\log_{k+1} |\mathcal{R}|)$ for both online and statistical settings (Theorem 9) in Appendix F.3.

### F.1    ONLINE LEARNER FOR pass@$k$ METRIC

We first start by describing an online learner in Algorithm 3.

---

**Algorithm 3** Online pass@$k$ rule with greedy selection and mistake unaware updates

---

**Input:** A finite model class $\mathcal{R} \subseteq [0,1]^{(\mathcal{X} \times \mathcal{Y})}$, and parameter $k \in \mathbb{N}$.
- Initialize $w^{(1)}(r) = 1$ for all $r \in \mathcal{R}$.
- In every round, receiving $x_t$:
  1. For each $y \in \mathcal{Y}$, form the slice $A_y^t = \{r \in \mathcal{R} : y \in \sigma_r(x_t)\}$.
  2. *(Greedy top-$k$ selection).* Let $\mathcal{Y}_0 = \emptyset$. For $i = 1, 2, \ldots, k$ set

$$\widehat{y}_t^{(i)} \in \arg\max_{y \in \mathcal{Y} \setminus \mathcal{Y}_{i-1}} w^{(t)}\Big(A_y^t \setminus \bigcup_{z \in \mathcal{Y}_{i-1}} A_z^t\Big), \qquad \mathcal{Y}_i \leftarrow \mathcal{Y}_{i-1} \cup \{\widehat{y}_t^{(i)}\}.$$

  (Break ties arbitrarily.) Define $U_t := \bigcup_{i=1}^k A_{\widehat{y}_t^{(i)}}^t$.
  3. Output the $k$ labels $\widehat{y}_t^{(1)}, \ldots, \widehat{y}_t^{(k)}$.
  4. *(Weight update).* Upon receiving label $y_t$:

$$w^{(t+1)}(r) \leftarrow \begin{cases} 0\,, & \text{for all } r \notin A_{y_t}^t; \\ w^{(t)}(r) & \text{for all } r \in A_{y_t}^t \cap U_t; \\ (k+1)\, w^{(t)}(r), & \text{for all } r \in A_{y_t}^t \setminus U_t. \end{cases}$$

---

We now prove the guarantee for the online rule, which formalizes the first part of Theorem 7.

**Theorem 8** (Online pass@$k$ guarantee). *On any sequence $((x_t, y_t))_{t \in \mathbb{N}}$, for any $r \in \mathcal{R}$ such that $\forall_t \, y_t \in \sigma_r(x_t)$, Algorithm 3 makes at most $\log_{k+1} |\mathcal{R}|$ mistakes with respect to $r$ (i.e., rounds with $\forall_{i \in [k]} \, \widehat{y}_t^{(i)} \notin \sigma_r(x_t)$ or equivalently $\{\widehat{y}_t^{(1)}, \ldots, \widehat{y}_t^{(k)}\} \cap \sigma_r(x_t) = \emptyset$).*

*Proof.* Define the potential $W_t := \sum_{r \in \mathcal{R}} w^{(t)}(r)$, then we again have $\{W_t\}_t$ is non-increasing.

$$
\begin{aligned}
W_{t+1} = (k+1) w^{(t)}(A_{y_t}^t \setminus U_t) + w^{(t)}(A_{y_t}^t) &= k \, w^{(t)}(A_{y_t}^t \setminus U_t) + w^{(t)}(A_{y_t}^t \setminus U_t) + w^{(t)}(A_{y_t}^t \cap U_t) \\
&\leq w^{(t)}(U_t \setminus A_{y_t}^t) + w^{(t)}(A_{y_t}^t \setminus U_t) + w^{(t)}(A_{y_t}^t \cap U_t) \\
&= w^{(t)}(U_t \cup A_{y_t}^t) \leq W_t,
\end{aligned}
$$

where in the first inequality arises from the following key relation:

$$
w^{(t)}(U_t \setminus A_{y_t}^t) \geq k \, w^{(t)}(A_{y_t}^t \setminus U_t). \tag{39}
$$

We defer its proof to the end of this section.

Now suppose the algorithm makes $M_r$ pass@$k$ mistakes (as defined in Theorem 8) by the end of round $t$ w.r.t. $r$. On each mistake round we must have $\sigma_\star \in A_{y_t}^t \setminus U_t$, so its weight is multiplied by $(k+1)$. Therefore

$$
w^{(t+1)}(\sigma_\star) = (k+1)^M \leq W_{t+1} \leq W_1 = |\mathcal{R}|,
$$

which yields $M \leq \log_{k+1} |\mathcal{R}|$. $\qquad\square$

**Proof of the key inequality (39).** To simplify the notation, we remove the time index $t$ and write $U_t$ as $U$, and $A_{y_t}^t$ as $A_{y_t}$. Define the uncovered mass in $A$ after selecting first $i$ labels greedily as:

$$
a_i := w^{(t)}\Big(A_{y_t} \setminus \bigcup_{z \in \mathcal{Y}_i} A_z\Big) \quad \text{so that} \quad a_0 = w^{(t)}(A_{y_t}), \;\; a_k = w^{(t)}(A_{y_t} \setminus U),
$$

and $a_0 \geq a_1 \geq \cdots \geq a_k$. Correspondingly, define

$$
s_i := a_{i-1} - a_i = w^{(t)}\Big(\big(A_{\widehat{y}_t^{(i)}} \cap A_{y_t}\big) \setminus \bigcup_{z \in \mathcal{Y}_{i-1}} A_z\Big).
$$

In addition, we define the uncovered weight for which $\widehat{y}_t^{(i)}$ got picked as

$$
m_i := w^{(t)}\Big(A_{\widehat{y}_t^{(i)}} \setminus \bigcup_{z \in \mathcal{Y}_{i-1}} A_z\Big).
$$

By the design of the greedy selections, we have

$$
m_i \geq a_{i-1} \qquad \text{for all } i \in [k]. \tag{40}
$$

With these definitions and relations in place, we are ready to prove the desired inequality. Basic set inclusion / exclusion tells us that

$$
\begin{aligned}
w^{(t)}(U \setminus A_{y_t}) &= \sum_{i=1}^k w^{(t)}(A_{\widehat{y}_t^{(i)}} \setminus A_{y_t} \cup A_{\widehat{y}_t^{(1)}} \cdots \cup A_{\widehat{y}_t^{(i-1)}}) \\
&= \sum_{i=1}^k \left\{ w^{(t)}(A_{\widehat{y}_t^{(i)}} \setminus \bigcup_{z \in \mathcal{Y}_{i-1}} A_z) - w^{(t)}\Big(\big(A_{\widehat{y}_t^{(i)}} \cap A_{y_t}\big) \setminus \bigcup_{z \in \mathcal{Y}_{i-1}} A_z\Big) \right\} \\
&= \sum_{i=1}^k (m_i - s_i),
\end{aligned}
$$

where the last equality uses the definitions of $m_i$ and $s_i$. Using (40), we obtain

$$
m_i - s_i \geq a_{i-1} - (a_{i-1} - a_i) = a_i,
$$

which allows us to further lower bound $w^{(t)}(U \setminus A)$ as

$$
w^{(t)}(U \setminus A_{y_t}) \geq \sum_{i=1}^k a_i \geq k \, a_k.
$$

Here we use the fact that $(a_i)_i$ is non-increasing. This proves the claim.

### F.2 STATISTICAL UPPER BOUND

We do the same online-to-batch conversion from Algorithm 2, based on a random stopping time of the online rule in Algorithm 3. We now prove the statistical guarantee for this estimator, to complete the proof of Theorem 7.

*Proof of Theorem 7.* We start by defining some notation. Let $\mu : \mathcal{X} \to \Delta(\mathcal{Y}^k)$ be any policy for the pass@$k$ metric. For any reward function $r : \mathcal{X} \times \mathcal{Y} \to [0, 1]$ and a distribution over prompt $\mathcal{D}$, we define the 0-1 loss of the policy $\mu$ as follows.

$$L_{\mathcal{D},r}(\mu) := \mathbb{E}_{x \sim \mathcal{D}} \mathbb{E}_{\boldsymbol{y} = (y^{(1)}, \dots, y^{(k)}) \sim \mu(\cdot | x)} \left[ \mathbb{1}\{ \forall i \in [k], y^{(i)} \notin \sigma_r(x) \} \right] .$$

**A reduction of reward maximization to loss minimization for an optimal demonstrator.** We now show that achieving small loss $L_{\mathcal{D},r}$ suffices to ensure low value suboptimality w.r.t. $r$. By definition:

$$
\begin{aligned}
V_r(\mu) &= \mathbb{E}_{x \sim \mathcal{D}} \mathbb{E}_{\boldsymbol{y} = (y^{(1)}, \dots, y^{(k)}) \sim \mu(\cdot | x)} \left[ \max_{1 \le i \le k} r(x, y^{(i)}) \right] \\
&\ge \mathbb{E}_{x \sim \mathcal{D}} \mathbb{E}_{\boldsymbol{y} = (y^{(1)}, \dots, y^{(k)}) \sim \mu(\cdot | x)} \left[ \mathbb{1}\{ \exists i \in [k], y^{(i)} \in \sigma_r(x) \} \max_{1 \le i \le k} r(x, y^{(i)}) \right] \\
&= \mathbb{E}_{x \sim \mathcal{D}} \left[ \sup_{y' \in \mathcal{Y}} r(x, y') \mathbb{E}_{\boldsymbol{y} = (y^{(1)}, \dots, y^{(k)}) \sim \mu(\cdot | x)} [\mathbb{1}\{ \exists i \in [k], y^{(i)} \in \sigma_r(x) \}] \right] \\
&= \mathbb{E}_{x \sim \mathcal{D}} \left[ \sup_{y' \in \mathcal{Y}} r(x, y') \left( 1 - \mathbb{P}_{\boldsymbol{y} = (y^{(1)}, \dots, y^{(k)}) \sim \mu(\cdot | x)} \left[ \forall i \in [k], y^{(i)} \notin \sigma_r(x) \right] \right) \right] \\
&\ge \mathbb{E}_{x \sim \mathcal{D}} \left[ \sup_{y' \in \mathcal{Y}} r(x, y') \right] - \mathbb{E}_{x \sim \mathcal{D}} \left[ \mathbb{P}_{\boldsymbol{y} = (y^{(1)}, \dots, y^{(k)}) \sim \mu(\cdot | x)} \left( \forall i \in [k], y^{(i)} \notin \sigma_r(x) \right) \right] \\
&\qquad\qquad\qquad\qquad\qquad\qquad\qquad\qquad\qquad\qquad\qquad\qquad\qquad\qquad\text{(since } r_\star(x, y') \le 1\text{)} \\
&= V_r^* - L_{\mathcal{D},r}(\mu) .
\end{aligned}
$$

This implies

$$V_r^* - V_r(\mu) \le L_{\mathcal{D},r}(\mu), \tag{41}$$

and thus, it suffices to show that the learner achieves low loss, which we do now.

**The estimator achieves low loss.** Let $\widehat{\mu}_{\mathrm{o2b}}$ be the output policy and $\{\widehat{\mu}_t : t = 1, \dots, m\}$ be the policies used in different rounds by the online learner. Consider the natural filtration $\mathcal{F}_t$ given by the first $t$ samples, and let $\mathbb{E}_t[\cdot] = \mathbb{E}[\cdot \mid \mathcal{F}_t]$ be the expectation conditioned on it. Let

$$\ell_t^r = \mathbb{1}\{ \forall i \in [k], \widehat{y}_t^{(i)}(x_t) \notin \sigma_r(x_t) \} .$$

Because $\widehat{\mu}_t$ is a deterministic function of $S_{<t} = \{(x_i, y_i) : i < t\}$, we have

$$\mathbb{E}_{t-1}[\ell_t^r] = \mathbb{E}[\ell_t^r \mid \mathcal{F}_{<t}] = L_{\mathcal{D},r}(\widehat{\mu}_t) .$$

We define the martingale difference sequence

$$Z_t^r := L_{\mathcal{D},r}(\widehat{\mu}_t) - \ell_t^r, \qquad \text{where } |Z_t| \le 1 \text{ almost surely.}$$

Then $\mathbb{E}_{t-1}[Z_t^r] = 0$, and

$$\mathbb{E}_{t-1}[(Z_t^r)^2] = \mathbb{E}_{t-1}\big[(L_{\mathcal{D},r}(\widehat{\mu}_t) - \ell_t^r)^2\big] = \mathrm{Var}_{t-1}(\ell_t^r) \le \mathbb{E}_{t-1}(\ell_t^r) = L_{\mathcal{D},r}(\widehat{\mu}_t) .$$

Applying Freedman's inequality (Lemma 3) with $R = 1$ and confidence parameter $\delta$, for any $\eta \in (0, 1)$ we have, with probability at least $1 - \delta$,

$$\sum_{t=1}^m Z_t^r \ \le \ \eta \sum_{t=1}^m L_{\mathcal{D},r}(\widehat{\pi}_t) + \frac{\log(2\delta^{-1})}{\eta} .$$

Substituting $Z_t^r = L_{\mathcal{D},r}(\widehat{\mu}_t) - \ell_t^r$ and rearranging gives

$$(1-\eta) \sum_{t=1}^{m} L_{\mathcal{D},r}(\widehat{\mu}_t) \ \leq \ \eta \sum_{t=1}^{m} \ell_t^r + \frac{\log(2\delta^{-1})}{\eta}.$$

By Theorem 8, we have that for any $r \in \mathcal{R}$, for which $\forall_{t\in[m]}, y_t \in \sigma_r(x_t)$, we have $\sum_{t=1}^{m} \ell_t^r \leq \log_{k+1} |\mathcal{R}|$. In particular for $r = r_\star$. This yields with probability at least $1 - \delta$

$$(1-\eta) \sum_{t=1}^{m} L_{\mathcal{D},r_\star}(\widehat{\mu}_t) \ \leq \ \eta \log_{k+1} |\mathcal{R}| + \frac{\log(2\delta^{-1})}{\eta}.$$

Substituting $\eta = 1/2$, and dividing both sides by $m$ gives us

$$\frac{1}{m} \sum_{t=1}^{m} L_{\mathcal{D},r_\star}(\widehat{\mu}_t) \ \leq \ \frac{\log_{k+1} |\mathcal{R}| + 4\log(2\delta^{-1})}{m}.$$

Finally, noting that the final averaged policy $\widehat{\mu}_{\mathrm{o2b}}$ is defined so that

$$L_{\mathcal{D},r_\star}(\widehat{\mu}_{\mathrm{o2b}}) = \frac{1}{m} \sum_{t=1}^{m} L_{\mathcal{D},r_\star}(\widehat{\mu}_t),$$

yields with probability at least $1 - \delta$

$$V_{r_\star}^* - V_{r_\star}(\widehat{\mu}_{\mathrm{o2b}}) \leq L_{\mathcal{D},r_\star}(\widehat{\mu}_{\mathrm{o2b}}) = \frac{1}{m} \sum_{t=1}^{m} L_{\mathcal{D},r_\star}(\widehat{\mu}_t) \leq \frac{\log_{k+1} |\mathcal{R}| + 4\log(2\delta^{-1})}{m},$$

here the first inequality follows from the equivalence derived earlier in Eq.(41). $\qquad\square$

### F.3   LOWER BOUNDS FOR ONLINE AND STATISTICAL SETTINGS FOR pass@$k$- METRIC

We next provide a lower bound that, information-theoretically, this dependence cannot be improved and we only gain a factor of $1/\log k$ in sample complexity as well as mistake bound in the worst-case. The lower bound instance shown here holds for even the simple special case of binary rewards and multiclass classification, so a single correct answer.

**Theorem 9** (Online $\Omega(\log_{k+1} |\mathcal{R}|)$ pass@$k$ mistake bound even for multiclass classification). *Fix integers $k \geq 1$ and $d \geq 2$. There exists a problem instance $\mathcal{R} \subseteq \mathcal{Y}^{\mathcal{X}}$ with $|\mathcal{R}| \leq d, |\mathcal{Y}| = k+1, |\mathcal{X}| = \lfloor \log_{k+1} d \rfloor$ such that for any deterministic online learning algorithm that outputs at most $k$ labels, there exists a sequence $(x_t, y_t)_{t\in[|\mathcal{X}|]}$ realizable by some $r_\star \in \mathcal{R}$ such that it makes mistake on every round.*

Note that our instance is an instance of multiclass classification problem $r : \mathcal{X} \to \mathcal{Y}$. This is isomorphic to an instance $\mathcal{R} \subseteq \{0,1\}^{\mathcal{X}\times\mathcal{Y}}$, where $r(x,y) = 1$ for only a single $y$ for every $x \in \mathcal{X}$.

*Proof of Theorem 9.* Let $m := \lfloor \log_{k+1} d \rfloor$ and take $\mathcal{X} = \{1, \ldots, m\}$ and $\mathcal{Y} = \{1, \ldots, k+1\}$. Consider the full product class $\mathcal{R} = \mathcal{Y}^{\mathcal{X}}$, which has size $(k+1)^m \leq d$. First of all, observe that in any round in which $y_t$ does not belong to the list of $(\widehat{y}_t^{(1)}, \ldots, \widehat{y}_t^{(k)})$, the mistake is made because we are in the multiclass classification setting.

For rounds $t \in [m]$, present a fresh coordinate $x_t = t$. Since $|\mathcal{Y}| = k+1$, there exists a label $y_t \in \mathcal{Y}$ that the learner failed to output in the set; $y_t \neq \widehat{y}_t^{(i)}$ for all $i \in [k]$. Reveal this $y_t$. This forces a mistake on every round. Moreover, this sequence is realizable since $\mathcal{R} = \mathcal{Y}^{\mathcal{X}}$ contains all functions from $\mathcal{X}$ to $\mathcal{Y}$. $\qquad\square$

We now prove a lower bound for the statistical setting. For a function $r : \mathcal{X} \to \mathcal{Y}$, we write $r(x)$ for the unique label it outputs on input $x$. Let $\mathcal{D} \times r$ denote the joint distribution where $\mathcal{D}$ is a distribution over contexts and $y \mid x = r(x)$ identically.

**Theorem 10** (Statistical lower bound of $\Omega(\log_k |\mathcal{R}|)$). *Fix integers $k \geq 1$ and $q \geq 1$. Let $\mathcal{X} = \{1, \ldots, q\}$, $\mathcal{Y} = \{1, \ldots, 2k\}$, and take the hypothesis class $\mathcal{R} = \mathcal{Y}^{\mathcal{X}}$ (all multiclass functions), so its cardinality is $d := |\mathcal{R}| = (2k)^q$. Let $\mathcal{D}$ be the uniform distribution on $\mathcal{X}$. Then for any estimators $\widehat{\mu} : (\mathcal{X} \times \mathcal{Y})^* \to \Delta(\mathcal{Y}^k)^{\mathcal{X}}$*

$$\inf_{\widehat{\mu}} \sup_{r \in \mathcal{R}} \mathbb{E}_{S \sim (\mathcal{D} \times r)^m} \mathbb{P}_{x \sim \mathcal{D}} \mathbb{P}_{\widehat{y}(x) \sim \widehat{\mu}(\cdot | x)} \big[ r(x) \notin \widehat{y}(x) \big] \geq \tfrac{1}{2} \Big( 1 - \tfrac{1}{q} \Big)^m.$$

*In particular, to ensure expected error at most $0 < \varepsilon < \frac{1}{2}$ for all $r \in \mathcal{R}$, one needs*

$$m \geq \frac{\ln(1/(2\varepsilon))}{-\ln(1 - 1/q)} \geq q \ln\Big(\frac{1}{2\varepsilon}\Big).$$

*Proof.* Fix any (possibly randomized) estimator $\widehat{\mu}$. Let $S = \{(x_i, y_i)\}_{i=1}^m$ be the training sample drawn i.i.d. from $(\mathcal{D} \times r)$ for $r \sim \mathrm{Unif}(\mathcal{R})$, and let $U_S = \{x_i : 1 \leq i \leq m\} \subseteq \mathcal{X}$ be the set of distinct inputs seen in $S$. Draw $x \sim \mathcal{D}$ independently of $S$ and then $\widehat{y}(x) \sim \widehat{\mu}(\cdot | x)$.

On any $x \notin U_S$, under the prior where $r$ is uniform over $\mathcal{R}$, for any (possibly randomized) $k$-list $\widehat{y}(x) \sim \widehat{\mu}(\cdot | x)$,

$$\mathbb{P}_r \big[ r(x) \in \widehat{y}(x) \mid S, x \notin U_S \big] = \mathbb{E}\left[ \frac{\# \text{ of distinct labels in } \widehat{y}(x)}{|\mathcal{Y}|} \,\bigg|\, S, x \notin U_S \right] \leq \frac{k}{2k} = \frac{1}{2},$$

so $\mathbb{P}_r[r(x) \notin \widehat{y}(x) \mid S, x \notin U_S] \geq \frac{1}{2}$. (Allowing duplicates in $\widehat{y}(x)$ cannot decrease this probability.)

If $x \in U_S$, the learner can always include the observed label and incur zero error on that $x$. Therefore, for any estimator $\widehat{\mu}$,

$$\mathbb{P}_{r, x, \widehat{y}(x) \sim \widehat{\mu}(\cdot | x)} \big[ r(x) \notin \widehat{y}(x) \mid S \big] \geq \frac{1}{2} \cdot \mathbb{P}[x \notin U_S].$$

Taking expectation over $S$ and using $\mathcal{D} = \mathrm{Unif}(\mathcal{X})$ yields

$$\mathbb{E}_S \, \mathbb{P}_{r, x, \widehat{y}(x) \sim \widehat{\mu}(\cdot | x)} \big[ r(x) \notin \widehat{y}(x) \big] \geq \frac{1}{2} \mathbb{E}_S \big[ 1 - |U_S|/q \big] = \frac{1}{2} \Big( 1 - \frac{1}{q} \Big)^m,$$

since $\mathbb{E}[|U_S|] = q\big(1 - (1 - \frac{1}{q})^m\big)$. Finally, by minimax principle

$$\inf_{\widehat{\mu}} \sup_{r \in \mathcal{R}} \mathbb{E}_S \, \mathbb{P}_{x \sim \mathcal{D}} \mathbb{P}_{\widehat{y}(x) \sim \widehat{\mu}(\cdot | x)} \big[ r(x) \notin \widehat{y}(x) \big] \geq \inf_{\widehat{\mu}} \mathbb{E}_{r \sim \mathrm{Unif}(\mathcal{R})} \mathbb{E}_S \, \mathbb{P}_{x \sim \mathcal{D}} \mathbb{P}_{\widehat{y}(x) \sim \widehat{\mu}(\cdot | x)} \big[ r(x) \notin \widehat{y}(x) \big]$$

$$\geq \tfrac{1}{2} \Big( 1 - \tfrac{1}{q} \Big)^m.$$

For the sample-complexity bound, solve $\frac{1}{2}(1 - \frac{1}{q})^m \leq \varepsilon$ for $m$ and use $-\ln(1 - 1/q) \leq 1/q$. $\square$

Because $d = (2k)^q$, we have $q = \log_{2k} d$, so the bound implies $m = \Omega\big(\log_k d\big)$ under $\mathcal{D} = \mathrm{Unif}(\mathcal{X})$.

**Remark 5.** Both online (Theorem 9) and statistical lower bounds (Theorem 10) for pass@$k$ essentially demonstrate that one cannot do better than memorization below $\Omega(\log_k d)$ barrier in the worst-case, even for the special case of the problem of realizable multiclass classification $\mathcal{R} \subseteq \{0, 1\}^{\mathcal{X} \times \mathcal{Y}}$ with a single correct answer and always correct demonstrations.

