# OpenReview forum: "Learning to Answer from Correct Demonstrations"
_ICLR.cc/2026/Conference — ICLR 2026 Poster_

### Official Review · Reviewer_aCdu · 2025-10-31

**Soundness:** 2
**Presentation:** 4
**Contribution:** 2
**Rating:** 2
**Confidence:** 3

**Summary:**

This paper addresses learning to generate answers to questions where multiple correct responses exist. The authors frame this as an offline imitation learning problem using training demonstrations, similar to supervised fine-tuning. Unlike previous approaches that assume demonstrators follow simple policies (justifying maximum likelihood methods), the authors make a weaker assumption: only the reward model (determining answer correctness) needs to be simple. They demonstrate that standard likelihood maximization can fail under this assumption. Their key contribution is a new learning algorithm with sample complexity that scales logarithmically with the number of possible reward functions. The method works robustly even with imperfect demonstrators and in relaxed evaluation settings (pass@k). The work suggests moving beyond likelihood maximization for learning from demonstrations may be beneficial.

**Strengths:**

1. The fundamental motivation, that a learner needs to produce a single correct answer and does not need to exactly mimic the demonstrator's correct answer, is a good one
2. The mathematical analysis carried out in the paper is extensive and rigorous
3. The paper is well-written and clear in its ideas, methods, and analysis

**Weaknesses:**

1. The reliance on reward model classes being finite and reward being binary is quite restrictive and limits the applicability of the theory of the method to tasks which are fuzzier and outside of domains such as maths. Even in programming one might want more nuanced reward signals (e.g. % of unit tests passed + some reward on how well formatted / structured the code is).
2. For domains with large state spaces (e.g. long context language modelling, often found in agentic tasks), the methods may fail to be performant due to their dependence on the size of the state space
3. No empirical experiments are carried out, not even for toy tasks.
4. It's not clear algorithm 1 is feasible to implement, since it takes in as inputs distributions over binary reward functions, and involve arg-maxing over outputs with terms inside the arg-max summing over huge spaces of reward functions. Outside of problems with few contexts and responses (in the contextual bandit sense), the algorithm is likely computationally intractable. This does not support the claim that your learner is in any way "efficient".
5. Critiques of the MLE paradigm are not supported by experiments or analysis of experimental data. Given the ability of LLMs to produce reasonable outputs (even if not perfect) after SFT training, this points to the mathematical framing of the paper being flawed, since some of its conclusions directly contradict the current state of affairs.
6. The learning algorithm is claimed to be logarithmic on $|S|$, however, given $S \subseteq (2^{\mathcal{Y}})^{\mathcal{X}}$, this makes the learner's convergence time scale linearly with the product of the context and output spaces in the worst case. Given the domain of interest is LLM finetuning, where these spaces are token sequences, the learner has an exponential dependence on the lengths of the input and output sequences. It is disingenuous to classify this as "efficient".

**Questions:**

1. Why did you not carry out any empirical experiments?
2. Are you able to actually implement your algorithm in code?
3. How do you square your theoretical work predicting the failure of MLE, when in practice it appears to work reasonably well (if not perfectly)?

---

> ### Author Response · Authors · 2025-11-24
> **Response**
>
> We thank the reviewer for their time in reviewing our work and bringing up concerns.
>
> ----
> **Regarding the reward class being binary (weakness 1):** As discussed in a comment to all reviewers, we now have a clean generalization for general bounded rewards.  This allows capturing partial correctness, or that some answers are better than others.
>
> **Regarding finiteness of the class (weakness 1):** We do have sample complexity dependence on the log-cardinality, and thus, strictly speaking, applicability only to finite classes, as significantly restrictive.  We view log-cardinality as a generic complexity measure that (crudely) captures the number of parameters: with 32-bit parameters, the log complexity of a model with N parameters is 32N.  Other similar analyses we relate to (e.g., Foster et al., Cohen et al.) also discuss learning in terms of log-cardinality.  Strictly speaking the analysis applies only to finite classes, but: (a) as alluded to earlier, model classes as represented on a computer are finite, and although their cardinality can be huge, their log-cardinality is proportional to the number of parameters and is quite reasonable; (b) more formally, instead of a finite class one can consider an $\epsilon$-cover of a continuous class, and rely on the size of this finite cover plus the $\epsilon$ approximation error.  It is fairly straightforward to state our results in terms of $L_\infty$ covering numbers of infinite classes, instead of cardinality, but we feel this is an unnecessary complication that does not provide additional insight and would stand in the way of the main message.  As you correctly point out, the main issue here is not with the sample complexity depending on the log-cardinality, but with the runtime and memory depending on the cardinality itself, and thus exponential in the number of parameters, and entirely impractical.
>
> ---
> **Regarding the claim of “efficiency” mentioned in weaknesses 4 & 6, and the dependence on the state space size mentioned in weaknesses 2 & 6:** we explicitly refer to **“sample-efficiency”**, i.e., using a small number of samples (question-answer pairs), and **not to computational efficiency**.  This is common statistical terminology, and we are careful to always be explicit about this—every time we use the word “efficient” in the paper, we say “sample-efficient”.  If you have suggestions to make this even more explicit, we will happily do so.
>
> **To emphasize again: what we claim, stand by, and view as entirely valid, is that the number of samples (i.e., number of demonstrations) required for learning depends only on the log-cardinality of the model class (roughly, the number of parameters, as discussed above) and does NOT depend on the size of prompt or response spaces nor the length of the prompt or response.**  (To realize that the log-cardinality of the model class doesn’t depend on the length of the input, consider, e.g., an encoder-only transformer taking x and y as input and outputting the reward---the log-cardinality corresponds to the number of parameters of the transformer, which is independent of the input length).  This is an extremely important point, and we would be happy to clarify this further if there is still a concern here. We do not claim any computational efficiency, and indeed, the runtime of the method, as stated, will be exponential in the output sequence length. The runtime and memory limitations are acknowledged in lines 495-501 (referencing the updated version), but we will be even more explicit to avoid any confusion.
>
> Our emphasis on statistical efficiency and sample complexity, setting aside the (important) issue of computational complexity for further investigation, is an important research approach in machine learning.  From the seminal work of Vapnik on Empirical Risk Minimization and Regularization, which completely ignored computational considerations but had profound practical implications later on, to many modern studies of reinforcement learning, active learning, adversarially robust learning, and domain adaptation that likewise studied ideas, limits, and plausible approaches both with and often without computational constraints.

---

> ### Author Response · Authors · 2025-11-24
> **Continuation**
>
> **Answering questions 1 and 2:**  Yes, it is very easy to implement our algorithm in code with a runtime that is linear in the number of possible reward functions and the number of possible responses.  Alternatively, with a runtime linear in the number of possible reward functions, and relying on a maximization oracle over responses.  The problem is not coding it up.  The problem, as you point out, is running it, since the runtime and memory would be prohibitive for any plausible model class.  For this reason, we did not carry out experiments.  We could carry out tiny synthetic experiments on constructed data, but the theory already tells us what would happen there.  The interesting question, as you point out, is to understand whether the gaps we prove exist in some problems are also manifested in actual SFT problems. We are yet unable to do so because our method is not yet practical for the scale of SFT, but we pointed out some conceptual connections earlier. Our work right now remains theoretical.
>
> ---
>
> **Regarding the critique of MLE (weakness 5 and question 3):**  We agree with you that MLE is “somewhat working depending on the situation, but not perfectly”. However, we respectfully disagree that there are any flaws with the mathematical phrasing of the reward class when learning from demonstrations.
> Our theoretical work establishes that there exist simple situations where it is possible to learn to answer correctly from demonstrations, but where MLE will fail to do so.  This is not to say that MLE will always fail---if we have demonstrations that are indeed coming from a generative demonstrator that is in a (small enough and known to us) demonstrator class, Proposition 1 ensures the success of MLE.  The question is whether this is the practice case, or whether this restrictive assumption does not hold, and other approaches might be better.  In some situations, it could be enough.  But as you also allude to, MLE is not perfect. To our knowledge, many SFT problems where MLE on the SFT data is often not sufficient, and post-SFT reward-based (RL) methods are used.
>
> Here, we are formally arguing that there are situations in which MLE is not the right choice, correcting a possible impression of optimality suggested by Foster et al. (see lines 68-78).  Also, under our reward class view, specifically in Remark 3.1, we suggest that there is some value in MLE, as, at the very least, it can achieve a non-trivial overlap with correct responses; in settings with only a few correct answers, this might be sufficient to produce a correct response with a decent probability.
>
> For this reason, we are not calling to “move beyond MLE” (as you mentioned in the review summary) but rather to “look beyond MLE” (line 24 of the abstract).  The distinction here is important: it is not that MLE never works and should be abandoned.  Rather, we shouldn’t focus just on MLE and develop alternate methods that could succeed where MLE fails when learning from demonstrations. To our knowledge, there have also been some purely practical fine-tuning methods (e.g., [1,2]) beyond MLE. Rather than trying to clone, these methods are self-discriminating the responses generated from the current or base model policy with the demonstrator policy (based on some explicit or implicit reward class). If we look at our Algorithm 1, then the algorithm also discriminates between the observed response $y_t$ and the generated response from its own current policy $\widehat{y}_t$. While these practical works set out to solve more engineering and practical issues and demonstrate empirical success, to our knowledge, theoretical components in these works do not address the core issue of sample efficiency of such methods in learning. The precise comparison with practical methods is out of scope yet, from the theory; here we are formally arguing why, for SFT data, there may be situations where MLE may not learn or have limited success in terms of overlap, but the methods with the right inductive biases can learn from such demonstrations sample-efficiently. We believe the conceptual advancement made in these works has direct relevance for SFT.
>
> ----
> ​​[1] Chen, Zixiang; Deng, Yihe; Yuan, Huizhuo; Ji, Kaixuan; Gu, Quanquan. (2024) Self-Play Fine-Tuning Converts Weak Language Models to Strong Language Models. In Proceedings of ICML 2024.
>
> [2] Li, Jiaxiang; Zeng, Siliang; Wai, Hoi-To; Li, Chenliang; Garcia, Alfredo; Hong, Mingyi. (2024). Getting More Juice Out of the SFT Data: Reward Learning from Human Demonstration Improves SFT for LLM Alignment. In Proceedings of NeurIPS 2024

---

### Official Review · Reviewer_Cuid · 2025-10-31

**Soundness:** 3
**Presentation:** 2
**Contribution:** 3
**Rating:** 6
**Confidence:** 5

**Summary:**

The paper studies the problem of learning from demonstrations when several correct outputs may exist for the same input. The authors argue that maximum likelihood estimation (MLE) fails in such cases because it imitates demonstrator distributions rather than learning correctness. They formalize this as an imitation learning problem with a low-cardinality reward class rather than a low-capacity policy class. Under this assumption, they show that MLE can overfit and fail to generalize, and propose an alternative “mistake-unaware” weighted-majority learner that provably converges to the correct reward mapping with logarithmic sample complexity. The theoretical results extend to non-optimal demonstrators and pass@k metrics

**Strengths:**

1. The paper tackles a relevant and well-motivated problem. The gap it identifies in how MLE behaves when multiple valid answers exist is real and interesting.

2. The proposed solution is conceptually neat: shifting from reasoning about small policy classes to small reward classes gives a different and insightful perspective on imitation learning.

3. The theoretical development is rigorous and internally consistent, with clear proofs and sound use of classical online-learning tools such as weighted-majority and halving arguments.

4. The exposition of the core idea, which distinguishes learning correctness from imitating distributions, is well articulated.

5. The paper effectively demonstrates that MLE’s failure is not statistical noise but a structural limitation, and that an alternative learner can overcome it in theory.

**Weaknesses:**

1. The work is entirely theoretical, with no experiments or simulations to illustrate MLE’s failure or the proposed learner’s behavior. This limits how convincingly one can assess its practical value.

2. The learner assumes a finite discrete hypothesis class and realizability (σ* ∈ S). While the authors acknowledge this, there is little discussion on what these assumptions imply for larger or continuous models, like LLMs.

3. The computational demands of maintaining weights over all hypotheses are briefly noted but not discussed in depth; in realistic settings, this would be infeasible.

4. While the motivation from LLMs is compelling, the theoretical framework (finite S, symbolic correctness sets) cannot directly model text generation. The connection to practical SFT or reward-learning pipelines remains purely conceptual.

5. The introduction could be more focused: it introduces several formal terms early on, which could instead appear in the problem-definition section. Presenting the motivation, gap, and main contributions upfront would make it easier to follow.

6. Although the proposed approach closes a theoretical gap, it remains unclear how it might inspire a usable method in practice. A short discussion of potential approximations or practical directions would help strengthen the paper.

**Questions:**

1. Could the authors comment on whether variants of their learner, perhaps approximate or sampling-based, might make it computationally feasible in larger hypothesis spaces?

2. Are there simple synthetic or illustrative experiments that could highlight the key behaviors (e.g. when MLE fails versus when the proposed learner succeeds)? Even a toy example might help ground the theory.

3. The paper assumes a finite hypothesis class; how sensitive are the results to mild relaxations of this assumption?

4. How might the “low-cardinality reward class” assumption be instantiated or approximated in realistic domains like LLMs?

---

> ### Author Response · Authors · 2025-11-24
> **Response**
>
> We thank the reviewer for their review and suggestions on our work!
>
> ---
> **Regarding weaknesses 2 (finiteness of the reward class) and 4 (symbolic correctness sets), and answering questions 3 and 4:**
>
> We attempted to discuss this in 486—501 (now in the updated version), but we now see our discussion can be improved.
>
> Since we want to emphasize cloning vs competing and MLE vs other approaches, we carry out our analysis in terms of the simplest complexity measure for learning, namely the log-cardinality.  This is a generic complexity measure and in a sense also (crudely) captures the complexity of parametric classes: with 32-bit parameters, the log cardinality of a class with N parameters is 32N.  Indeed, other similar analyses we relate to (e.g., Foster et al, Cohen et al.) also discuss learning in terms of log-cardinality.  Strictly speaking the analysis applies only to finite classes, but: (a) as alluded to earlier, model classes as represented on a computer are finite, and although their cardinality can be huge, their log-cardinality is proportional to the number of parameters and is quite reasonable; (b) more formally, instead of a finite class one can consider an $\epsilon$-cover of a continuous class, and rely on the size of this finite cover plus the epsilon approximation error.  It is fairly straight-forward to state our results in terms of $L_\infty$ covering numbers of infinite classes, instead of cardinality, but we feel this is an unnecessary complication that does not provide additional insight and would stand in the way of the main message; (c) from a mathematical perspective, it would be nice to study truly continuous classes, and obtain guarantees in terms of measures like generalization of the VC dimension, and/or study specific classes like transformers and understand their complexity (though even the VC dimension of transformers is not yet fully understood!).
>
> In terms of correctness classes relevant to LLMs and “symbolic correctness sets”: what we envision as the reward class is an encoder-only transformer that takes as input the prompt x and the candidate answer y, and outputs the “correctness” or reward.  We find it fairly compelling to view correctness this way (either binary correctness, or especially following our update, a softer fractional notion).  In particular, if you have a next-token-generator transformer T, you can construct a verifier transformer T’ such that T’(x,y)=1 iff y is a high-probability generation by T with prompt x.  And so, whatever notion can be captured by generation with a transformer, can also be captured by verification with a transformer.  Again, log-cardinality should be thought of as “number of parameters in the transformer”.
>
> The main problem, as we see it, and as we discuss in lines 496—497 (cf. updated version), is not that the sample complexity relies on the log-cardinality (this we find satisfying), but that in our current approach runtime and memory are linear in the cardinality, and so exponential in the number of parameters.  But in the same way that for supervised learning and RL, many generic sample complexity results ignore these (important!) computational issues, and motivate development of appropriate tractable and practical methods, we are hoping that our statistical analysis will likely serve as such a motivator.
>
> We will improve the discussion in the paper to better communicate these points, and are very happy to answer additional questions regarding this. We indeed require realizability.  But since we do not require the demonstrator to be optimal (see the common response), this is less severe.
>
> ----
>
> We now answer the questions in order.
>
> 1. Practical Variants: unfortunately, at this point we do not have a practical variant. We believe that understanding continuous classes (see lines 486-500 cf. updated version) with linear scores or even generalized linear score models is an important next step, needed to lead to a practical algorithm. Maintaining weights over discretized space and approximate sampling may not be the correct way to proceed. Algorithms that directly work with parameterization (e.g.,time polynomial in number of parameters) may be desirable.
> 2. Because of the above limitations, unfortunately, we don’t currently see any practical implementation of the algorithm , which limits us with the possibility of running any interesting experiments for question-answering in LLMs.
> 3. Sensitivity to finiteness: See discussion above.
> 4. See discussion above.

---

### Official Review · Reviewer_FNKo · 2025-11-02

**Soundness:** 4
**Presentation:** 3
**Contribution:** 4
**Rating:** 8
**Confidence:** 4

**Summary:**

This paper studies the problem of learning to generate answers (or completions) to questions (or prompts) when there can be multiple correct answers, based on demonstrations of correct answers. Compared to prior work, the main novelty of this paper is that it relies on assumptions that **the reward model lies in a low-cardinality class**, rather than the assumption that **the demonstrator belongs to a low-complexity policy class**.

Specifically, this paper formulates the problem in Section 2 and discusses why MLE can fail under this problem formulation in Section 3. Novel learning algorithms are proposed and analyzed in Section 4. Various possible extensions are discussed in Section 5. Concluding remarks and possible future work are discussed in Section 6.

**Strengths:**

Overall, I think this is a thought-provoking and solid theoretical paper. Specifically,

- This paper is motivated by very practical problems in LLMs, i.e., many QA problems in math/coding can have many equally valid but differently written solutions. It has also developed a rigorous contextual bandit framework to analyze them. More importantly, it has identified major limitations of MLE under this framework. As a researcher in the field of RL/bandit and LLM, I find this paper very thought-provoking.

- The learning algorithms proposed in this paper are interesting.

- The proposed learning algorithms, and some of their extensions, have been extensively analyzed. To the best of my knowledge, the analyses are rigorous and correct.

Overall, I think this is a strong paper and recommend accepting it.

**Weaknesses:**

- One obvious limitation of this paper is that it has not provided any experimental results. Given that this paper is motivated by practical problems in LLMs, I recommend that the authors include some experimental results on LLMs in this paper. I think the experimental results will show the following:
  - how well the proposed algorithm will perform in **practical** LLM settings, compared to MLE-based algorithms;
  - the computational tractability/efficiency of the proposed algorithms in **practical** LLM settings.
These are hard to tell from mathematical analyses.

**Questions:**

- Please add some experimental results if possible.

- [Minor] I think the "learning from arbitrary demonstrator" setting (Section 5.2) is a more general and more interesting setting. It is not clear to me why the authors don't present that setting as the main problem formulation of this paper, and discuss the case when $\pi_*$ is optimal as a special case. I am wondering why.

---

> ### Author Response · Authors · 2025-11-24
> **Response**
>
> We thank the reviewer for positive feedback on the work and suggestions!
>
> To answer your 2nd minor question: indeed, the setup of learning from an arbitrary demonstrator is more general, and we now aim to present it exactly the way you suggested—see the common response to all reviewers for some updates.
>
> Experiments: Unfortunately, the algorithm as presented is not practical, as it requires maintaining weights on every possible reward function, as well as searching over all possible responses (as acknowledged in lines 495-501 now in the updated version), which limits us with the possibility of running interesting experiments for LLMs at this time.  At this point, we view the paper similarly to how you stated it: as a thought-provoking theoretical paper, where we tried to emphasize crisp and clean results.  We are working on practical implementations motivated by this, and hope the paper will motivate others to do the same.

---

### Official Review · Reviewer_mbR1 · 2025-11-06

**Soundness:** 4
**Presentation:** 4
**Contribution:** 3
**Rating:** 8
**Confidence:** 3

**Summary:**

This paper studies the offline imitation learning problem in contextual bandits, under the model that only correct demonstrations are provided. This problem is of practical relevance for the supervised training of LLMs. Departing from prior work, the authors do not assume access to a small policy class (to model the conditional distribution of responses given prompts), but only the weaker assumption of access to a small "reward" class, assigning a binary reward of 1 to responses that are correct and 0 to responses that are incorrect. It is shown that standard algorithms (i.e., MLE) can fail under this weaker assumption, and new algorithms are devised that learn under this relaxed reward class assumption. Several extensions are provided.

**Strengths:**

The shift from policy classes to reward classes is well-motivated and potentially practically interesting. The presentation is very clean. The technical results are interesting and check out (at a glance). The results establish that MLE fails and introduce new algorithmic ideas for imitation learning from reward classes, and these are tight in terms of their dependence on the log-cardinality class. The authors study several extensions of interest, such as non-optimal demonstrators and pass@k metrics.

**Weaknesses:**

I don't have many complaints. The main drawbacks include: 1. the analysis assumes a finite reward class, and since the algorithm is Majority/EXP-style this seems non-trivial to extend, 2. the algorithm requires enumeration over S and is thus not computationally implementable. It would be highly desirable to obtain a more computationally tractable algorithm.

**Questions:**

- Do you envision convenient parameterizations of the reward class S (e.g. linear classifiers) where efficient implementations are possible?

---

> ### Author Response · Authors · 2025-11-24
> **Response**
>
> Thank you for positive feedback on the work!
>
>
> To answer the questions: understanding continuous parametrized classes is indeed an important next question.  We do think linearly parametrized classes, as discussed in lines 488—494 (now in the updated version) could be approachable.  With general bounded rewards (see common response), the appropriate generalization is perhaps $\mathcal{R} = \\{ r_u(x,y) = g( \langle u , \phi(x,y) \rangle) \\}$.  We do think the inclusion of a fixed non-linearity $g:\mathbb{R}\rightarrow [0,1]$ is important here.  This can be approached either by approximating $w(r_u)$ in the algorithm with weight functions $w(r_u)$ that have a simple parametric form in terms of $u$, e.g., perhaps only quadratics in $u$ (this is similar, in a sense, to an online ellipsoid algorithm that maintains version space specified by a quadratic, instead of a general polytope).  Or, by replacing the multiplicative weight updates with gradient steps on $u$ directly.  We do not yet know if this approach can enjoy strong theoretical guarantees.

---

### Author Response · Authors · 2025-11-24
**Common Response with Some Updates and Connections**

We thank all the reviewers for their time and positive feedback on our work!  In addition to addressing some reviewer concerns and questions, we would like to share an update on the paper with you.

---
Following submission, we obtained a clean generalization of our algorithm for **real-valued bounded reward** (not just binary rewards) and **without assuming the demonstrator is optimal**.  That is:
- We directly consider a general (finite) reward class $\mathcal{R}=\\{ r: \mathcal{X} \times \mathcal{Y} \rightarrow [0,1] \\}$ as in the extension in Section 5.1 of our original submission. This extends the special case we focused on throughout, where all the rewards in the class are binary and of the form $r(x,y) = \mathbf{1}\\{y \in \sigma(x)\\}$.
- And we can directly achieve $\varepsilon$-value suboptimality with respect to an arbitrary (not necessarily optimal) demonstrator for a general reward $r_* \in \mathcal{R}$. In the original version Extension 5.2, we just considered the binary-reward case when discussing suboptimal demonstrator, where the value of the demonstrator $V(\pi_{dem})=1-L_{\mathcal{D},\sigma_*}(\pi_{dem})$. With this new result of achieving $\varepsilon$-suboptimality, we now achieve loss-suboptimality with a coefficient of “1” on the demonstrator’s loss, instead of “1.41” as in Theorem 5 of the original submission. More importantly, we can compete with the value for general bounded "real valued" reward functions even when the demonstrator is "slightly suboptimal but never perfect" (so the loss according to optimal actions is 1, but the value can still be close to optimal).

Our bottom line guarantee is (Theorem 5 in the updated version): we can output a policy $\widehat{\pi}$ with value
$V(\widehat{\pi}) \geq V(\pi_{dem}) -  O\left( \sqrt{\frac{\Delta \log( |\mathcal{R}| /\delta )}{m}} + \frac{\log (|\mathcal{R}|/\delta)}{m} \right) $
where $\Delta \in [0,1]$ is a bound on suboptimality of the demonstrator $\Delta_{dem}= V^*-V(\pi_{dem})$.  We can always take $\Delta=1$ and obtain a $1/\sqrt{m}$ for competing with any arbitrary demonstrator, but this more refined bounds provides an “optimistic rate” interpolating between the $1/m$ rate when the demonstrator is optimal, and the worst case $1/\sqrt{m}$ which always holds. This formulation is completely in terms of the reward class, which subsumes the results about the support function class $\mathcal{S}$ as a special case in it.

This generalization is obtained by a generalization of the online Algorithm 1, updating the weight of each candidate reward function based on upweighting and downweighting reward hypotheses in terms of the suboptimalities of the **online algorithm's response** $\widehat{y}_t$ and the **demonstrator's response** $y_t$ (see new Algorithm 1 Eq 4, updated version). A direct online analysis (and online-to-batch conversion), with appropriate hyperparameters, yields the unified guarantee above, and completely subsumes extension in 5.1 and 5.2 of the original submission, and goes beyond it. (We still have the treatment of pass@k as an extension in Section 5.)

**Note on the updated PDF**:  We have made minimal necessary changes to help understand these general results for reviewers who have already read the submitted version.

---

> ### Author Response · Authors · 2025-11-24
> **Continued...**
>
> ----
> **Additional connections to prior and parallel work**: after obtaining the general form above, we realized a connection to Sayed and Schapire's “A Game-Theoretic Approach to Apprenticeship Learning” (NIPS 2007).  They discuss learning an MDP policy by observing a demonstrator, where rewards are an unknown bounded L1-norm linear function of known feature space.  Viewing each reward function as a feature, and replacing their treatment of discounted reward and the horizon with a treatment more appropriate for contextual bandits, it's possible to use their result (Theorem 2) to get a result very similar to ours, namely that with $m$ observations from a demonstrator $\pi_*$ we can learn a policy with value $V(\widehat{\pi}) \geq V(\pi_{dem}) - O( \sqrt{ \log(|\mathcal{R}|/\delta) / m })$.  On a technical level, the benefit of our approach over Sayed and Schapire is:
> - More direct, and arguably simpler, method and analysis
> - Optimistic rate, with a $1/m$ rate when the demonstrator is optimal (as opposed to $1/\sqrt{m}$ always
> - Adversarial online guarantee, not just stochastic. Even in the stochastic setting, demonstrations can be adaptive as long as they are good.
> - One-pass method, as opposed to Sayed and Schapire, which present a batch method requiring many passes over the demonstrations.
> - Thanks to its simplicity and directness, we can also apply our approach to pass@k.
>
> Beyond these technical differences, we want to emphasize that the contrast to MLE and cloning, presenting a small policy class vs a small reward class formulation for “Learning to Answer from Correct Demonstrations” in both Definition 1, and the Online version is distinct and different.
>
> After our submission, in a private communication with Moulin, Neu, and Viano, we were also informed of their work “Inverse $Q$-Learning Done Right: Offline Imitation Learning in $Q^{\pi}$-Realizable MDPs”, to appear in the upcoming NeurIPS 2025. This is also related to our main result.  They do not discuss contextual bandits, but their results can be invoked also for our setting, ensuring $V(\widehat{\pi}) \geq V(\pi_{dem}) - O(\sqrt[4]{ \log |\mathcal{R}| \log |\mathcal{Y}|/m})$, again using a multi-pass algorithm.  This is both a substantially worse rate and even more significantly involves a dependence on the size $\log|\mathcal{Y}|$ of the output space, which for prompt-response scales with the length of the responses, and which we emphasize avoiding.
>
> We will discuss these works in detail in the final version.
>
>
> ----
> [1] Syed, Umar; Schapire, Robert E. (2007). A Game-Theoretic Approach to Apprenticeship Learning. (NIPS 2007)
>
> [2] Moulin, Antoine; Neu, Gergely; Viano, Luca. (2025). Inverse $Q$-Learning Done Right: Offline Imitation Learning in $Q^{\pi}$-Realizable MDPs. To appear in NeurIPS 2025

---

### Author Response · Authors · 2025-12-03
**A short discussion summary for AC**

Dear AC,

Thank you for your service! We would like to summarize the discussion. Three of the reviewers had a very positive evaluation of our work.  Furthermore, we updated the paper addressing two points raised by the reviewers (addressing the finiteness requirement and generalizing the results significantly beyond binary rewards and an optimal demonstrator), which would have probably even further increased their appreciation of the paper.  A low score (Rating 2)  by Reviewer aCdu stems from some misunderstanding, which we clarified, but they did not have a chance to update (see more below).
- The reviewers (mbR1, Cuid, aCdu) asked about the finiteness assumption and relying on cardinality.  We explicitly addressed this in the responses and, following the reviews, also added an important paragraph to the paper introduction (lines 65-90) that explains how the log cardinality corresponds to the number of parameters in natural classes (e.g., rewards encoded via a transformer).  We hope this new paragraph helps better explain the generality of our results.
- Reviewers aCdu and FNKo had concerns about the restriction to binary rewards and the assumption that the demonstrator is optimal, suggesting a limitation of our analysis, as the suboptimal demonstrator case and general non-binary rewards, we only had in a limited way before.  Indeed, we agreed, and we have now significantly generalized our results and present a simple, general main result (Theorem 5 + Algorithm 1) that applies to both binary and real-valued rewards, without any assumption of optimality.
- Reviewer aCdu seems to have been confused between sample and computational efficiency and tried to interpret our results as claiming computational efficiency, which we certainly do not.  We are careful to always write “sample-efficient” (never simply “efficient”), and explicitly discuss the computational interactability of our method.  We believe our clarification would have helped the reviewer better appreciate our contribution and hopefully also increase their score.
- Another orthogonal issue Reviewer aCdu was concerned about is the failure of MLE under the new proposed reward class assumption, suggesting a contradiction since MLE somewhat works, but not perfectly, in practice. We clarified to them that even within our framework, such situations for MLE are captured through an "overlap with good answers", but the situations of its failure for the actual goal of "producing good answers" are laid out.  This is to correct the image of MLE as perfect and minimax optimal under the small Policy Class Assumption, from prior work. We do not view these failures as abandoning MLE (as the reviewers perhaps perceived the message), but rather also look beyond it (as we said in the abstract).

---

### Meta-Review · Area_Chair_G1Zr · 2025-12-09

**Summary:**

This paper studies offline imitation learning for contextual bandits in a setting relevant to supervised fine-tuning (SFT) of LLMs, focusing on the case where multiple correct responses may exist for each prompt. Rather than assuming a low-complexity policy class that is standard in prior work, the authors assume that the reward model lies in a low-cardinality class. Under this weaker assumption, they show that MLE could fail. They propose an alternative online weighted-majority style learner whose sample complexity scales only with the log cardinality of the reward class.

Most reviewers agree that the theoretical shift is clean and interesting, and the analysis is rigorous. The weaknesses mainly lie in the practical relevance, including finite reward classes, restriction to binary rewards, the issue of a computationally tractable algorithm, no experiments, and only a conceptual connection to LLMs.

The authors' rebuttal addressed several concerns. They generalized their results from binary to bounded real-valued rewards and both optimal and suboptimal demonstrators. For finiteness, they also added an explanation relating log-cardinality to parameter counts of realistic models. However, practical implementability and the lack of empirical validation remain unaddressed after rebuttal. The algorithm is still computationally infeasible in realistic settings, and the authors argue that experiments are not feasible at this stage.

The paper makes a clean and nontrivial conceptual contribution, backed by tight theoretical guarantees. Most reviewers support acceptance. Although the rebuttal did not address all the concerns, the issue of practical relevance could be acceptable for a theory paper. The authors are encouraged to further clarify sample vs computational efficiency, refine the discussion on how log-cardinality relates to realistic model classes, and add a more explicit section on potential practical approximations and future directions for tractability.

**Reviewer Concerns:**

Concerns addressed by the rebuttal:

1. Finiteness / log-cardinality assumption. (mbR1, Cuid, aCdu)
2. Binary reward limitation. (FNKo, aCdu)
3. Assumption of optimal demonstrator. (FNKo, aCdu)

Concerns still outstanding:

1. Computational infeasibility of the algorithm (mbR1, Cuid, aCdu, FNKo)
2. No empirical evidence (FNKo, Cuid, aCdu)
3. Realism of symbolic reward-class modeling for LLMs (Cuid)

**Reviewer Scores:**

Although the authors provided the rebuttal, some concerns remain. The scores are likely unchanged.

* Reviewer mbR1: 8
* Reviewer FNKo: 8
* Reviewer Cuid: 6
* Reviewer aCdu: 2

---

### Decision · Program_Chairs · 2026-01-26

Accept (Poster)